# Canonical versus non-canonical transsynaptic signaling of neuroligin 3 tunes development of sociality in mice

Neuroligin 3 (NLGN3) and neurexins (NRXNs) constitute a canonical transsynaptic cell-adhesion pair, which has been implicated in autism. In autism spectrum disorder (ASD) development of sociality can be impaired. However, the molecular mechanism underlying NLGN3-mediated social development is unclear. Here, we identify non-canonical interactions between NLGN3 and protein tyrosine phosphatase δ (PTPδ) splice variants, competing with NRXN binding. NLGN3-PTPδ complex structure revealed a splicing-dependent interaction mode and competition mechanism between PTPδ and NRXNs. Mice carrying a NLGN3 mutation that selectively impairs NLGN3-NRXN interaction show increased sociability, whereas mice where the NLGN3-PTPδ interaction is impaired exhibit impaired social behavior and enhanced motor learning, with imbalance in excitatory/inhibitory synaptic protein expressions, as reported in the *Nlgn3* R451C autism model. At neuronal level, the autism-related *Nlgn3* R451C mutation causes selective impairment in the non-canonical pathway. Our findings suggest that canonical and non-canonical NLGN3 pathways compete and regulate the development of sociality.

Social behaviors for properly communicating with others are mediated by various cortical and subcortical neural circuits, including the medial prefrontal cortex (mPFC), amygdala, anterior insula, anterior cingulate cortex, inferior frontal gyrus, and superior temporal sulcus. Developmental dysregulation of these social brain circuits is associated with the pathophysiology of autism spectrum disorders (ASDs), a group of neurodevelopmental disorders with core symptoms of impaired social interaction and communication accompanied by restricted interests and repetitive behaviors[1,2]. Genetic and genomic studies have revealed that distortions of transsynaptic signaling mediated by synaptic cell adhesion proteins are closely associated with the pathogenesis of ASDs[3–6]. The X-linked *neuroligin 3* (*NLGN3*) gene encodes the postsynaptic adhesion molecule NLGN3, which interacts with presynaptic neurexins (NRXNs) to organize synaptogenesis. *NLGN3* is one of the best-characterized sociality-related genes, with various mutations including deletions and an arginine-to-cysteine missense (R451C) mutation linked to ASDs[7–10]. *Nlgn3* knock-in mice harboring the autism-related R451C mutation (*Nlgn3*^{R/C}) and *Nlgn3* knockout (KO) mice display enhanced ability to remain on the accelerating rotarod associated with the repetitive behaviors of ASD[11]. In contrast, decreased social interaction accompanied by imbalances in the excitatory and inhibitory synaptic protein expression and transmissions in the cerebral cortex and hippocampus is observed exclusively in the *Nlgn3*^{R/C} mice, and not in the KO mice, even though the R451C mutation seems to be a loss-of-function mutation that causes reduced affinity to the canonical binding partners, NRXNs, and intracellular retention of the mutant protein in the endoplasmic reticulum[12–15]. Therefore, the molecular mechanism underlying the NLGN3-mediated social development, as well as the pathogenic mechanism of ASD-linked NLGN3 mutations, remains a mystery. In fact, intriguingly, NLGN3 KO and the R451C mutation differentially impact synaptic transmission in a circuit-, neuronal cell-, or synapse-type-specific manner, and the R451C mutation differentially affects two different types of input onto the same postsynaptic cell[11,16–18], implying that NLGN3 may utilize different types of transsynaptic signaling.

Mutations in *NRXN1*, one of the three members of the *NRXN* gene family, have also been detected in ASD-affected individuals, although they appear to predispose to a wider range of neurodevelopmental and psychiatric disorders, probably because NRXNs function as a presynaptic hub to interact with multifarious postsynaptic partners[6]. The canonical postsynaptic partners of NRXNs include NLGNs, leucine-rich repeat transmembrane neuronal proteins, and cerebellin precursor proteins-GluDs (ref. [6]). Type IIA receptor protein tyrosine phosphatases (RPTPs; PTPδ, PTPσ, and leukocyte common antigen-related (LAR) in mammals) are another group of presynaptic hub proteins[5]. Type IIA RPTPs exist in multiple isoforms generated by alternative splicing of microexons at the sites corresponding to a loop within the extracellular second immunoglobulin (Ig)-like domain (mini-exon A; meA) and the junction between the second and third Ig-like domains (mini-exon B; meB), with the greatest variation having been observed in PTPδ (Fig. 1a) (refs. [19,20]). The postsynaptic ligands for PTPδ include interleukin-1 receptor accessory protein (IL-1RAcP), IL-1RAcP-like-1 (IL1RAPL1), Slit- and Trk-like proteins (Slitrks), and synaptic adhesion-like molecules (SALMs), most of which prefer PTPδ splice variants containing the meB peptide[21–25].

Here, we identified a non-canonical synapse-organizing interaction between NLGN3 and PTPδ splice variants lacking the meB peptide, which competed with NRXNs by steric hindrance. The crystal structure of the NLGN3–PTPδ complex revealed the splicing-dependent interaction mechanism for synapse organization. Based on the atomic-level comparison of NLGN3–PTPδ and the reported NLGN1/4–NRXN1 interfaces, we designed *Nlgn3* knock-in mutations in mice to dissect the physiological roles of the canonical and non-canonical pathways. Our results suggest that a balance between the canonical and non-canonical NLGN3 pathways may contribute to the development of sociality, with the autism-related R451C mutation disrupting this balance to diminish the non-canonical PTPδ–NLGN3 pathway.

## Results

**PTPδ splice variants have distinct synaptogenic properties.** The developing mouse brain expresses at least eight Ig domain-splice variants of PTPδ (Fig. 1a)[20]. We examined the postsynapse-inducing activity of these PTPδ splice variants using an artificial synaptogenic assay. Cortical neurons were incubated with magnetic beads coated with equal amounts of the respective extracellular domains (ECDs) of the PTPδ splice variants, followed by immunostaining for the excitatory and inhibitory postsynaptic

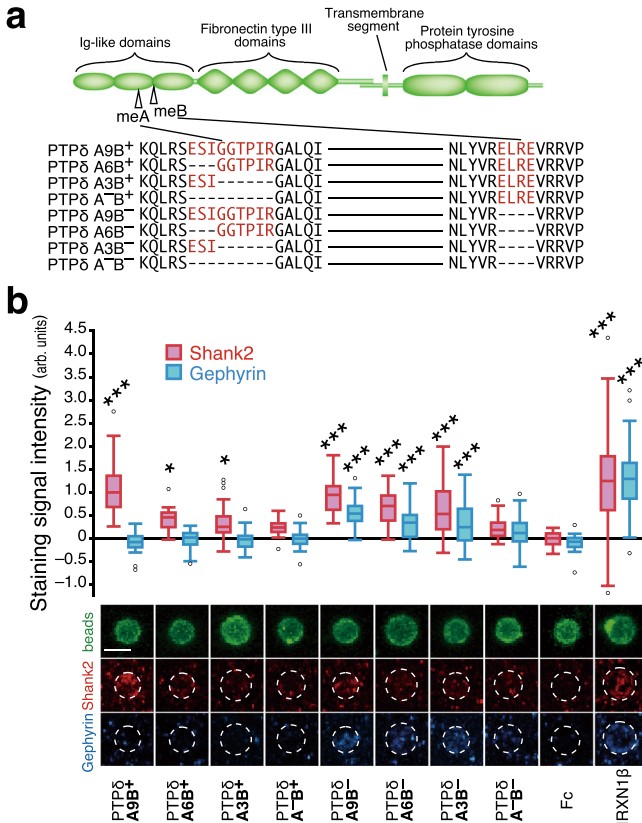

**Fig. 1 Properties of PTPδ splice variants to induce postsynaptic differentiation. a** Schematic structure of PTPδ and amino acid sequences of splice variants in Ig-like domains. me A and B sequences are colored in red. **b** Induction of postsynaptic differentiation of cerebral cortical neurons by beads conjugated with ECDs of PTPδ splice variants. Excitatory and inhibitory postsynaptic terminals were visualized by immunostaining for Shank2 (red) and gephyrin (blue), respectively (bottom). Shank2 (red bars) and gephyrin (blue bars) staining signals on the beads (n = 35, 35, 37, 38, 35, 35, 36, 35, 35, and 35 for PTPδ A9B⁺, A6B⁺, A3B⁺, A⁻B⁺, A9B⁻, A6B⁻, A3B⁻, A⁻B⁻, Fc, and NRXN1β beads, respectively) were quantified (top). Scale bar, 5 μm. Data are presented as box plots. Horizontal line in each box shows the median, the box shows the interquartile range (IQR) and the whiskers are 1.5 × IQR. *P < 0.05 and ***P < 0.001, Tukey's post hoc test compared to control Fc beads. See Supplementary Table 4 for additional statistics and exact p values.

scaffold proteins Shank2 and gephyrin, respectively (Fig. 1b). Selective accumulation of Shank2, but not of gephyrin, was detected on the cortical neurons in contact with the beads coated with meB-peptide-containing PTPδ variants (PTPδ-meB(+)s; PTPδA9B+, PTPδA6B+, PTPδA3B+, and PTPδA−B+). In contrast, accumulation of both Shank2 and gephyrin was detected on the beads coated with ECDs of PTPδ variants lacking the meB peptide (PTPδ-meB(−)s; PTPδA9B−, PTPδA6B−, PTPδA3B−, and PTPδA−B−). Moreover, the signal intensities for Shank2 and gephyrin were dependent on the size of the meA peptides, i.e., the longer the meA peptides, the greater the accumulation of the postsynaptic proteins on the beads. These results suggest that PTPδ-meB(+)s selectively induce the excitatory postsynaptic differentiation, whereas PTPδ-meB(−)s are involved in both excitatory and inhibitory postsynaptic differentiation.

**Identification of NLGN3 as a selective ligand for PTPδ-meB(−)s.** The differential postsynapse-inducing properties of PTPδ splice variants imply the existence of diverse postsynaptic ligands selective for each PTPδ splice variant. The previously characterized PTPδ ligands IL1RAPL1, IL-1RAcP, Slitrks, and SALMs, preferentially interact with PTPδ-meB(+)s[20–25]. In fact, the excitatory postsynaptic differentiation induced by PTPδA9B+, a PTPδ-meB(+) isoform abundantly expressed in the developing mouse brain[20], was reduced by ~70% in cultured cortical neurons from IL1RAPL1/IL-1RAcP double-knockout (DKO) mice (Supplementary Fig. 1a, b), suggesting IL1RAPL1 and IL-1RAcP are major postsynaptic ligands for PTPδA9B+ to induce excitatory synapse formation in the cortical neurons. In contrast, IL1RAPL1/IL-1RAcP DKO had little effect on the postsynaptic differentiation induced by PTPδA3B−, the most abundant PTPδ-meB(−) isoform (Supplementary Fig. 1c, d), suggesting that PTPδ-meB(−)s utilize other ligand(s) for post-synaptic differentiation.

To identify the ligands that mediate PTPδ-meB(−)-induced postsynaptic differentiation, we employed in situ proteomic screening (Fig. 2a)[26]. After postsynaptic differentiation of cultured cortical neurons was induced on beads conjugated with the ECD of PTPδA3B−, postsynaptic proteins that extracellularly interacted with PTPδA3B− were chemically cross-linked and recovered. Liquid chromatography-tandem mass spectrometry of the PTPδA3B− complex identified four transmembrane proteins, including NLGN3 (Supplementary Table 1).

We confirmed the direct interaction between PTPδ splice variants and NLGN3 using a cell surface binding (CSB) assay and a cell aggregation assay. Among the mouse NLGN3 splice variants containing (NLGN3A) or lacking (NLGN3(−)) the splice segment (SS) A in the ECD, the NLGN3A isoform was used in this study, and "NLGN3" indicates this isoform, henceforth. HEK293T cells expressing NLGN3 were incubated with the soluble ECDs of PTPδ splice variants fused to Fc and then stained with an anti-Fc antibody, to quantify the cell surface-bound Fc fusion proteins (Fig. 2b, c). We detected staining signals on cells incubated with the ECDs of PTPδ-meB(−)s, but not those of PTPδ-meB(+)s. Correspondingly, HEK293T cells expressing NLGN3 selectively aggregated with those expressing PTPδ-meB(−)s (Fig. 2d). NLGN3 also interacted with LAR and PTPσ splice variants lacking the meB peptide, although the interactions were much weaker than the interaction with PTPδ A3B− (Supplementary Fig. 2a). In contrast, among the four murine NLGN family members, NLGN3 splice variants alone showed significant binding to PTPδA3B− (Supplementary Fig. 2b), suggesting that PTPδ-meB(−)s interact exclusively with NLGN3. We further confirmed the NLGN3-PTPδ interaction in vivo by coimmunoprecipitation assay using anti-NLGN3 antibody in the synaptosomal fraction

of striatums where highest PtprdA3B− expression was observed (Fig. 2e and Supplementary Fig. 2c).

Next, we examined the involvement of NLGN3 in PTPδ-meB (−)-induced postsynaptic differentiation by coculture assays using cortical neurons from neuron-specific NLGN3 KO mice that lacked the exon 7 by Cre recombinase under the control of the nestin promoter (Supplementary Fig. 3a–c). Exon 7 encodes a half of the ECD, and skipping of this exon is predicted to lead to a frame shift and a premature stop codon. In NLGN3 KO neurons, no accumulation of Shank2 or gephyrin was detected on the beads coated with PTPδ-meB(−)s, indicating that PTPδ-meB(−)s require NLGN3 to induce postsynaptic differentiation (Fig. 2f, g, and Supplementary Fig. 3d–f). In contrast, NLGN3 KO had a negligible effect on the synaptogenic activities of PTPδ-meB(+)s. These results suggest that the selective interaction between NLGN3 and PTPδ-meB(−)s organizes both excitatory and inhibitory synaptogenesis.

**PTPδ and NRXN1 compete for binding to NLGN3.** The iden-tification of the non-canonical NLGN3–PTPδ-meB(−) interaction prompted us to examine its physiological and pathological roles in brain development for sociality by designing and analyzing NLGN3 mutants that dissect the canonical NRXN- and non-canonical PTPδ-meB(−)-mediated synapse-organizing pathways. Therefore, we examined the relationship between PTPδ-meB(−)s and NRXN1β for binding to NLGN3. An NRXN1β isoform lacking SS4 showing the strongest NLGN3 binding among the NRXN family members was used in this study[27]. In a CSB assay using soluble PTPδA3B−-Fc and NLGN3-expressing HEK293T cells, cell surface-bound PTPδA3B−-Fc signals were reduced by the addition of similar concentrations of hexahistidine-tagged PTPδA3B−-ECD or NRXN1β-ECD but not of alkaline phosphatase (Fig. 2h). Furthermore, preformed NLGN3-NRXN1β complex on the cell surface was replaced by the NLGN3-PTPδA3B− complex by adding PTPδA3B−-Fc (Sup-plementary Fig. 2d). Consistently, isothermal titration calori-metric (ITC) analyses showed the binding affinity between NLGN3 and PTPδA3B− ($K_D = 4.4\,\mu M$) is comparable to that between NLGN3 and NRXN1β ($K_D = 1.8\,\mu M$ in the presence of $1\,mM\ Ca^{2+}$) (Fig. 2i). Furthermore, the NLGN3 mutant with three alanine substitution mutations at Leu374, Asn375, and Asp377 in the putative interface with NRXN1β (NLGN3 LND)[28] failed to interact with PTPδA3B− (Supplementary Fig. 2e). These results suggest that PTPδA3B− and NRXN1β compete for binding to NLGN3 by sharing interfaces. This is further supported by the fact that, MDGA1, known as an NLGN3 interactor that inhibits NRXN interaction[29], also interfered with the NLGN3-PTPδ interaction in competitive CSB assay (Supplementary Fig. 2f). Therefore, atomic-level structural information at their interfaces is required to design genetic tools to separately dissect the canonical and non-canonical transsynaptic signaling pathways of NLGN3.

**Structural basis of the NLGN3–PTPδA3B− interaction.** To elucidate the structural mechanism of the NLGN3–PTPδ-meB(−) interaction, we determined the crystal structures of mouse apo-NLGN3 ECD (residues 35–684) at 2.76 Å resolution (Supple-mentary Table 2 and Supplementary Fig. 4a) and NLGN3 ECD in complex with PTPδA3B− Ig1–Fn1 at 3.85 Å resolution (Supple-mentary Table 2 and Fig. 3). The NLGN3 structure adopts an α/β-hydrolase fold, which is topologically identical to the reported structures of NLGN1, 2, and 4[28,30–32]. The electron densities corresponding to residues 148–173 and 555–567 were invisible, likely owing to the structural disorder. The first disordered region contains a 19-residue insertion at SS-A. The asymmetric unit of

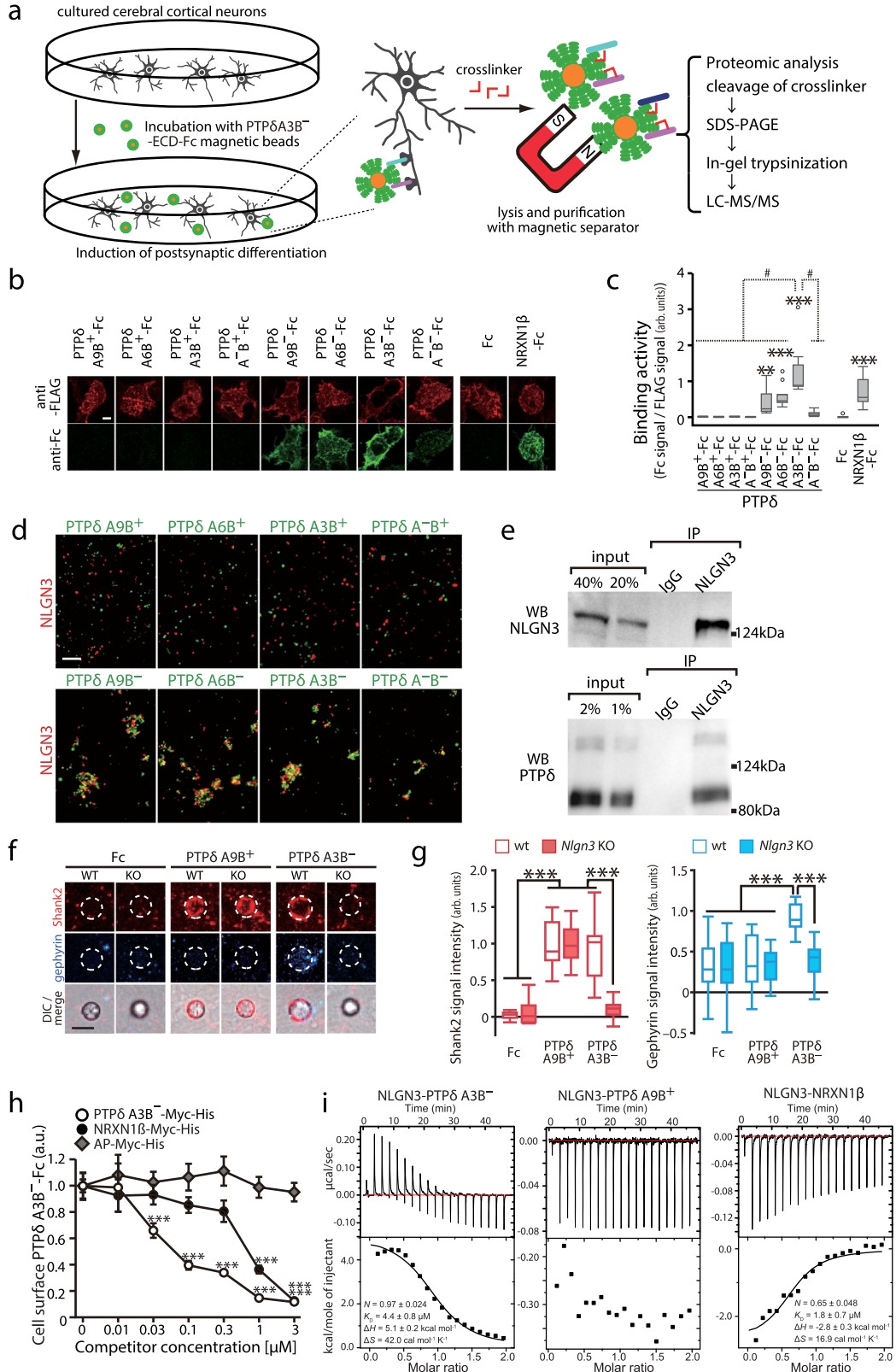

the apo-NLGN3 crystal contains two NLGN3 molecules, which form a dimer (Supplementary Fig. 4a). The molar mass of the recombinant mouse NLGN3 ECD determined by size-exclusion chromatography coupled with multiangle light scattering (SEC-MALS) analysis (Supplementary Fig. 4b) is 160 kDa, which corresponds to the mouse NLGN3 ECD dimer. All reported NLGN dimer structures, including our NLGN3 structure, exhibit a similar dimerization interface, where two helices from each protomer form a four-helix bundle[28,30–32]. In the crystal of the NLGN3–PTPδA3B− complex, two adjacent NLGN3 molecules that are related by twofold crystallographic symmetry form a dimer in a manner similar to apo-NLGN3 ECD. Consequently,

**Fig. 2 Identification and characterization of non-canonical interactions between NLGN3 and PTPδ splice variants lacking meB. a** Schema for in situ screening of postsynaptic ligands for PTPδA3B⁻ during hemi-synapse formation. **b** Cell surface binding of ECDs of PTPδ splice variants fused to Fc (green) to HEK293T cells expressing FLAG-NLGN3 (red). **c** Ratios of staining signals for Fc fusion proteins and FLAG-NLGN3 in (**b**) (*n* = 15 cells each). **d** Representative images of cell aggregation assay of HEK293T cells coexpressing NLGN3 and red fluorescent protein (red) and those coexpressing PTPδ splice variants and green fluorescent protein (green). The experiment was repeated independently twice or more. **e** Representative blots of coimmunoprecipitation (co-IP) assay of NLGN3 and PTPδ from mouse brain. The experiment was repeated independently twice. **f** PTPδA9B⁺ and PTPδA3B⁻ bead-induced excitatory and inhibitory postsynaptic differentiation of wild-type (WT) and NLGN3 knockout (KO) cerebral cortical neurons visualized by Shank2 (red) and gephyrin (blue) immunostaining, respectively. **g** Intensity of staining signals for Shank2 and gephyrin on the PTPδA9B⁺ and PTPδA3B⁻ beads in (**f**) (*n* = 17 and 12 Fc beads, 21 and 14 PTPδA9B⁺ beads, and 19 and 22 PTPδA3B⁻ beads for wild-type and *Nlgn3* KO neurons, respectively). **h** Competitive cell surface binding assay. Cell surface-bound PTPδA3B⁻-Fc signals in the presence of 0–3.0 µM alkaline phosphatase (AP)-Myc-His, PTPδA3B⁻-Myc-His or NRXN1β-Myc-His were quantified (*n* = 14 cells each). **i** ITC titration curves for binding of NLGN3 to PTPδ splicing variants and NRXN1β. Data in (**c**) and (**g**) are presented as box plots. The horizontal line in each box shows median, box shows the IQR and the whiskers are 1.5 × IQR. Data in (**h**) represent mean ± s.e.m. Scale bars, 5 µm. *$p < 0.05$, **$p < 0.01$, and ***$p < 0.001$, Tukey's test in (**g**) and (**h**), and two-sided Dunnett's test in (**c**) compared to Fc control. #$p < 0.001$, Tukey's test compared to PTPδA3B⁻-Fc in (**c**). See Supplementary Table 4 for additional statistics and exact *p* values.

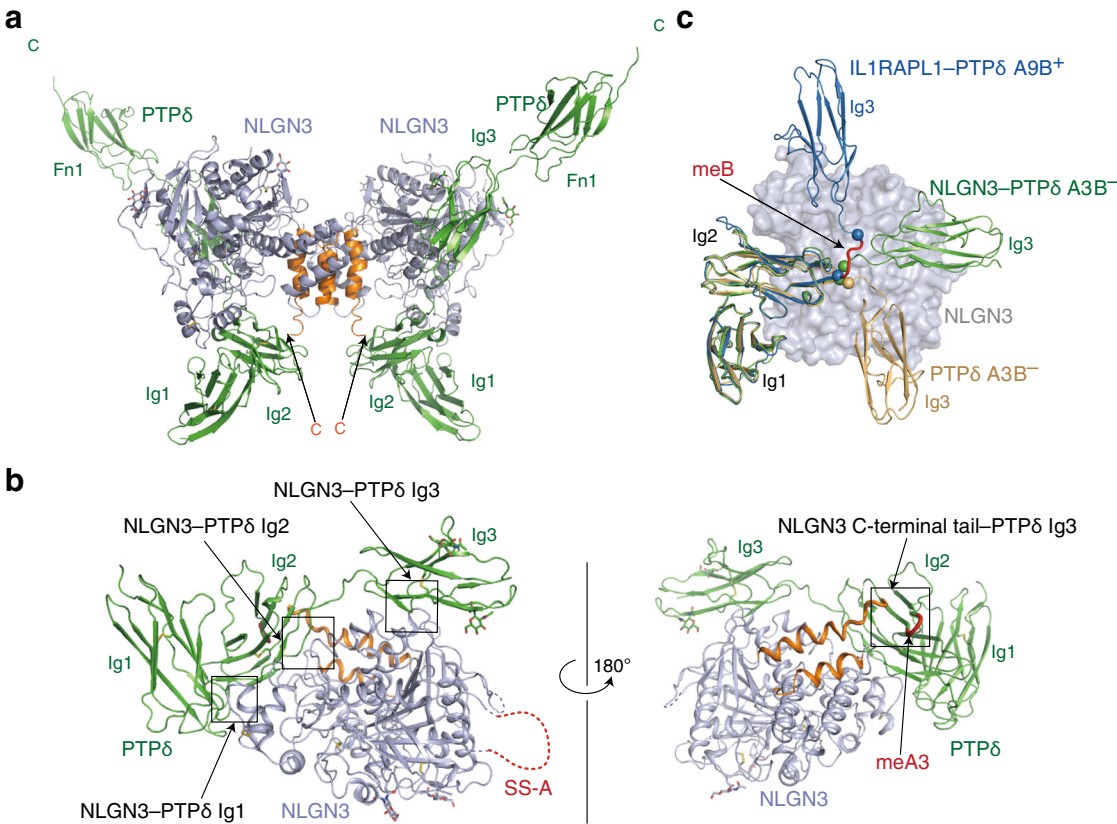

**Fig. 3 Structure of the NLGN3–PTPδA3B⁻ complex. a** Overall structure of the NLGN3 ECD–PTPδA3B⁻ Ig1–Fn1 heterotetrameric complex. Two NLGN3 molecules are purple, except that their C-terminal regions containing two dimerization helices are orange. Two PTPδ molecules are colored in green. **b** Binding interfaces between NLGN3 and PTPδA3B⁻. The individual interfaces are indicated by boxes. The coloring scheme is the same as that in (**a**), except that the meA3 insertion in PTPδ is red. PTPδ Fn1 is not shown for clarity. **c** Superposition of the NLGN3-bound PTPδA3B⁻ Ig1–Fn1 (green), apo-PTPδA3B⁻ Ig1–Fn2 (PDB 4YFG; beige), and IL1RAPL1-bound PTPδA9B⁺ (PDB 4YH6; blue) using PTPδ Ig1–Ig2 as the reference. The bound NLGN3 is shown as a molecular surface colored in purple. The Fn domains of PTPδ or the bound IL1RAPL1 are not shown for clarity. The meB insertion is red. The Cα atoms of PTPδA9B⁺ Arg233 and Val238, which flank the meB insertion, are shown as spheres. Arg227 and Val228 of PTPδA3B⁻ (equivalent to Arg233 and Val238 of PTPδA9B⁺, respectively) are also shown as spheres.

PTPδA3B⁻ and NLGN3 form a W-shaped complex with a 2:2 stoichiometry (Fig. 3a and Supplementary Fig. 5a).

The Ig1–Ig3 domains of PTPδ interact extensively with the α/β-hydrolase-fold core of NLGN3 (Fig. 3b, c). The Ig2 domain of PTPδ also contacts the C-terminal tail of NLGN3 ECD (Fig. 3b). The NLGN3–PTPδ interface buries a surface of 1540 Å² in total. Phe170 and the aliphatic portions of Arg75, Glu77, Arg90, and Gln92 in PTPδ Ig1 form a hydrophobic pocket to accommodate the side chain of Leu320 in NLGN3 (Fig. 4a and Supplementary

Fig. 5b). Leu141, Pro221, and Tyr225 of PTPδ Ig2 form a hydrophobic interface with Tyr302, Val305, and Ile355 in the α/β-hydrolase-fold core of NLGN3 (Fig. 4b and Supplementary Fig. 5c). In addition, the C-terminal tail of NLGN3 ECD lies on the meA3 insertion in PTPδ Ig2 (Fig. 4c and Supplementary Fig. 5d). The C-terminal tail of NLGN3 ECD is disordered in the apo-NLGN3 ECD structure, suggesting that it undergoes a disorder-to-order transition upon binding to PTPδ. Met614 and Phe615 of NLGN3 appear likely to form hydrophobic interactions

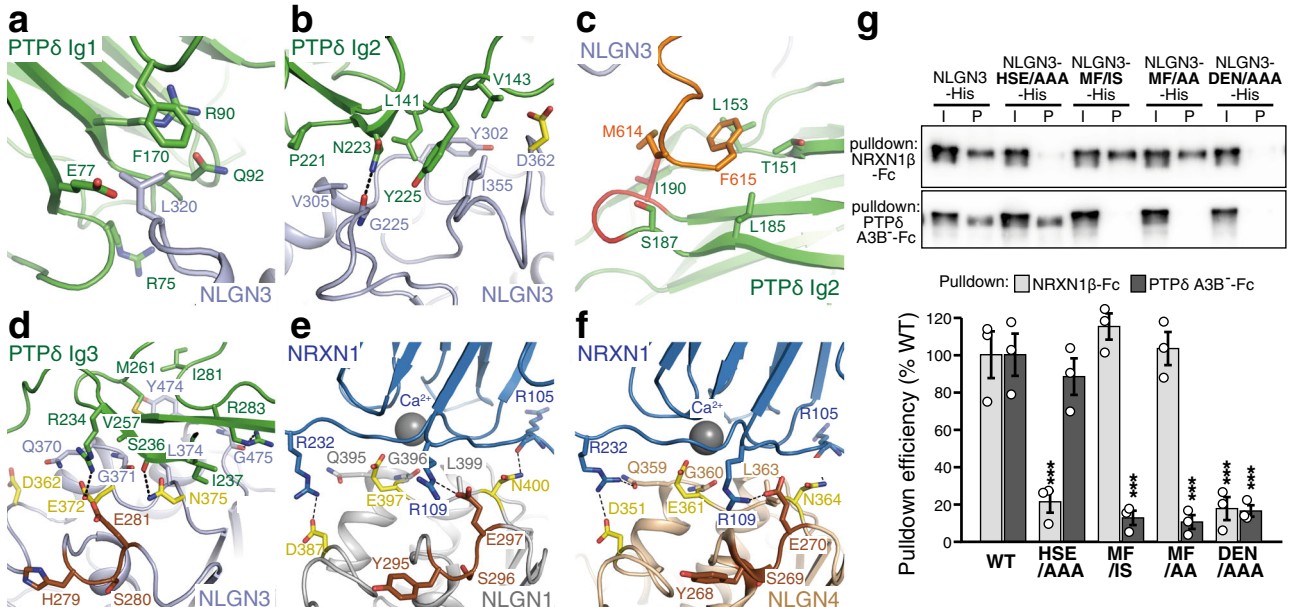

**Fig. 4 Binding interfaces between NLGN3 and PTPδ. a–c** Close-up views of the NLGN3 α/β-hydrolase-fold core/PTPδ Ig1 interface (**a**), NLGN3 α/β-hydrolase-fold core/PTPδ Ig2 interface (**b**), and NLGN3 ECD C-terminal/PTPδ Ig2 interface (**c**). The coloring scheme is the same as that in Fig. 3b. **d–f** Close-up views of the NLGN3/PTPδ Ig3 interface (**d**), NLGN1/NRXN1β (PDB 3B3Q) interface (**e**), and NLGN4/NRXN1β (PDB 2WQZ) interface (**f**). The coloring scheme is the same as that in (**a**), except that (putative) NRXN/PTPδ-interacting residues (Asp362, Glu372, and Asn375) and NRXN-interacting loop (His279, Ser280, and Glu281) of NLGN3 are yellow and brown, respectively. The bound calcium ion is shown as a gray sphere. The NRXN-interacting residues of NLGN1 (Asp387, Glu397, and Asn400) and NLGN4 (Asp351, Glu361, and Asn364) are yellow. Tyr295, Ser296, and Glu297 of NLGN1, and Tyr268, Ser269, and Glu270 of NLGN4, which correspond to the NRXN-interacting loop of NLGN3, are brown. **g** NRXN1β or PTPδA3B⁻ bound wild-type NLGN3, NLGN3-HSE/AAA, NLGN3-MF/IS, NLGN3-MF/AA, and NLGN3-DEN/AAA mutants with His₆ tag were resolved by SDS-PAGE and blotted with anti-His antibody (top). The pulldown efficiency of wild-type or mutated NLGN3 was quantified (bottom; $n = 3$ experiments). Data are mean ± s.e.m. ***$p <$ 0.001, Tukey's test. See Supplementary Table 4 for exact $p$ values.

with Leu153, Leu185, Ile190, and the aliphatic portions of Thr151 and Ser187 in PTPδ Ig2. Ile190 is included in the meA3 insertion. The contribution of the meA3 insertion to the binding is consistent with the results of our CSB and cell aggregation assays (Fig. 2b–d), which showed lower binding of PTPδA⁻B⁻ to NLGN3 compared with PTPδA3B⁻. The relative orientation of Ig3 to Ig2 in the NLGN3-bound PTPδA3B⁻ is totally different from that in the apo-PTPδA3B⁻ (ref. [24]): the Ig3 domain of PTPδ rotates by ~160 degrees upon binding to NLGN3 (Fig. 3c)[33]. Val257, Met261, and Ile281 of PTPδ Ig3 form a hydrophobic patch to interact with Gly371, Leu374, and Tyr474 of NLGN3 (Fig. 4d). Ser236 and Arg283 of PTPδ may form hydrogen bonds with Asn375 and the main chain of Gly475 in NLGN3, respectively. In addition, PTPδ Arg234 is positioned close enough to form a hydrogen bond with Glu372 of NLGN3. The domain rearrangement of PTPδ Ig3 allows both PTPδ Ig2 and Ig3 to simultaneously interact with NLGN3. The relative spacing between the Ig2 and Ig3 domains appears to be adjusted by the length of the meB-lacking linker connecting these two domains; the meB-containing linker is likely to be more flexible with an extended spacing (9 Å longer than the meB-lacking linker), as deduced from the comparison with the IL1RAPL1- and IL-1RAcP-bound PTPδA9B⁺ (Fig. 3c)[24].

**Dissection of the NLGN3–PTPδ and NLGN3–NRXN pathways.** The amino-acid sequences of the NRXN-binding site are highly conserved among all NLGNs including NLGN3 (Supplementary Fig. 4c). The (potential) NRXN-binding site of NLGN3 comprises His279–Glu281, Asp362, and Gln370–Asp377 (Fig. 4d), which correspond to Tyr295–Glu297, Asp387, and Gln395–Asp402 of NLGN1 (Fig. 4e) or Tyr268–Glu270, Asp351, and Gln359–Asp366 of NLGN4 (Fig. 4f), respectively.

The binding site of NLGN3 for PTPδ Ig3 overlaps with the (potential) NRXN-binding site. The bound PTPδ Ig3 covers Gln370–Asp377 (Fig. 4d). Specifically, Gly371, Glu372, Leu374, and Asn375 of NLGN3 interact directly with PTPδ Ig3. Moreover, Asp362 of NLGN3, which is equivalent to Asp387 of NLGN1, contacts Val143 of PTPδ in close proximity to the PTPδ Ig2–NLGN3 interface (Fig. 4b). These structural features are consistent with the results of the abovementioned cell surface-binding assay of NLGN3 LND (Supplementary Fig. 2e) and the ITC experiments mentioned below.

Based on the crystal structures of the NLGN–NRXN and NLGN3–PTPδ complexes, we designed three distinct types of NLGN3 mutants to dissect the NRXN- and PTPδ-meB (−)-mediated pathways. The first type is the H279A S280A E281A triple mutation (hereafter referred to as HSE/AAA). His279, Ser380, and Glu281, which comprise a loop in the proximity of the common interaction site for PTPδ and NRXNs, appear to contribute to binding to NRXNs (Fig. 4d) but not to PTPδ. NLGN1 Glu297 and NLGN4 Glu270, which are equivalent to NLGN3 Glu281, form a hydrogen bond with NRXN1β Arg109 (Fig. 4e, f)[34,35], which is conserved in NRXN2 and NRXN3. In fact, the NLGN3 HSE/AAA mutation abolished the interaction with all NRXNs without affecting the affinity to PTPδA3B⁻ ($K_D$ = 1.4 μM) (Figs. 2i, 4g, and Supplementary Fig. 6c–e). The second group of mutants contains the M614I F615S or M614A F615A double mutation (hereafter referred to as MF/IS or MF/AA, respectively), which disturbs the interface between PTPδ Ig2 and the C-terminal tail of NLGN3 ECD without any effect on the interaction with NRXNs. In fact, PTPδA3B⁻ failed to pull down the MF/IS and MF/AA mutants of NLGN3 (Fig. 4g). The $K_D$ of the NLGN3 MF/IS mutant for PTPδA3B⁻ was 27 μM, which is 6 times lower than that of wild-type NLGN3, whereas that for

NRXN1β was 2.3 μM, which is comparable to the affinity of wild-type NLGN3 (Fig. 2i and Supplementary Fig. 6a). The MF/AA mutation more severely blocked the PTPδA3B⁻ interaction than the MF/IS in the CSB assay (Supplementary Fig. 6e). The third is the D362A E372A N375A triple mutation (hereafter referred to as DEN/AAA), which disturbs a common interaction site for PTPδ and NRXNs. No binding of the NLGN3 DEN/AAA mutant to NRXN1β or PTPδA3B⁻ was detected in the pulldown and ITC experiments (Fig. 4g and Supplementary Fig. 6b, e). Altogether, we successfully obtained NLGN3 mutants with different binding selectivity for PTPδA3B⁻ and NRXN1β as potential molecular tools for dissecting the canonical and non-canonical transsynaptic signaling of NLGN3.

**Differential effects on sociality by blockade of the canonical and non-canonical NLGN3 signaling.** To address the physiological roles of the canonical NLGN3–NRXN and non-canonical NLGN3–PTPδ pathways in social development, we generated knock-in mutant mice on a pure C57BL/6 genetic background with the HSE/AAA mutation (*Nlgn3ʰˢᵉ* line) and MF/AA mutation (*Nlgn3ᵐᶠ* line), respectively, using CRISPR/Cas9 genome editing (Supplementary Fig. 7). The HSE/AAA and MF/AA mutations did not alter the stability of the NLGN3 protein itself in the mouse brain (Supplementary Fig. 7d, g). We confirmed the impairment of the canonical and non-canonical synaptogenic pathways in these mutant mice by coculture assays using NRXN1β and PTPδA3B⁻ beads, respectively. These mutations did not seem to change background staining signals for Shank2 and gephyrin in cultured neurons (Supplementary Fig. 7h, i). In cortical neurons from *Nlgn3ʰˢᵉ* male mice, NRXN1β-beads failed to induce accumulation of NLGN3 while Shank2 and gephyrin signals on the NRXN1β-beads were attenuated but not completely abolished (Fig. 5a, c). In contrast, the PTPδA3B⁻-mediated accumulation of NLGN3 and gephyrin, but not Shank2, was rather upregulated (Fig. 5a, c). The residual postsynapse-inducing activity of NRXN1β beads in the *Nlgn3ʰˢᵉ* cortical neurons was probably mediated by other NRXN1β ligands such as NLGN1, NLGN2, NLGN4 and LRRTMs. In cortical neurons from *Nlgn3ᵐᶠ* male mice, PTPδA3B⁻ beads failed to induce NLGN3, Shank2, and gephyrin accumulation, while NRXN1β-induced NLGN3 and gephyrin accumulation was markedly enhanced (Fig. 5b, d). Consistently, anti-NLGN3 antibody failed to coimmunoprecipitate PTPδ in the *Nlgn3ᵐᶠ* mice brain, whereas it coimmunoprecipitated more PTPδ in the brains of the *Nlgn3ʰˢᵉ* mutants than in those of wild-type littermates (Fig. 5e). These results suggest that the canonical NLGN3–NRXN and non-canonical NLGN3–PTPδ transsynaptic pathways are actually impaired in the *Nlgn3ʰˢᵉ* and *Nlgn3ᵐᶠ* lines, respectively. Furthermore, the canonical and non-canonical pathways seem to counterbalance each other for inhibitory synapse formation.

Next, the development of social behaviors in these mutant mice was assessed using the three-chamber social approach test[36]. In this test, a wire cage with a stranger mouse was placed in one side chamber and an empty cage was placed in the other side chamber. Both *Nlgn3ʰˢᵉ* mutants and their wild-type littermates spent more time in the chamber with the stranger mouse than in the chamber with the non-social empty cage (Fig. 5f). However, unexpectedly, *Nlgn3ʰˢᵉ* mutants spent significantly more time in the chamber with the stranger (Fig. 5f) and more preferentially stayed around the stranger cage than the littermate controls (Fig. 5g and Supplementary Fig. 8a–c, g), suggesting that disruption of the canonical NLGN3–NRXN interaction enhances sociability development. In contrast, in *Nlgn3ᵐᶠ* mutant mice impairing the non-canonical PTPδ-pathway, the time spent in the chamber with the stranger mouse was not significantly longer

than that with non-social empty cage, whereas their wild-type littermates spent more time in the chamber with the stranger (Fig. 5h), suggesting no obvious enhancement of social preference in the *Nlgn3ᵐᶠ* mutants. Although the social-behavioral phenotype of *Nlgn3ᵐᶠ* mice was not robust, *Nlgn3ᵐᶠ* mice showed significantly shorter duration in each stranger-cage-sniffing behavior than their wild-type littermates (Fig. 5h,i and Supplementary Fig. 8d–f, h). To further characterize the social behavior of the mutant mice, the reciprocal social interaction was recorded and analyzed using a 3D-video-based markerless motion capture system (Fig. 5j–n)[37]. In the test, both *Nlgn3ʰˢᵉ* and *Nlgn3ᵐᶠ* mutant mice spent more time in proximity with the other mice than their wild-type littermates (Fig. 5j, l and Supplementary Fig. 8i, j). We then examined a 'defensive upright posture'[38] by measuring elevated head-head contacts during the test. Both *Nlgn3ʰˢᵉ* and their littermate wild-type mice showed comparable numbers of the elevated head-head contacts throughout the test period while *Nlgn3ᵐᶠ* mice exhibited the contacts more frequently than their wild-type littermates in the middle of the test session (Fig. 5m, n). These results suggest both *Nlgn3ʰˢᵉ* and *Nlgn3ᵐᶠ* mutants showed increased social interaction but their qualities seem different.

Regarding ASD-related behavioral phenotypes other than social abnormalities, enhanced motor routine learning is observed in *Nlgn3* KO and *Nlgn3ᴿ/ᶜ* mice[11]. We therefore subjected our mutant mice to the accelerating rotarod test, in which the rotation speed increased up to 80 rpm over 300 s (Fig. 5o, p). *Nlgn3ᵐᶠ* mice showed a significantly higher performance than wild-type littermates in this accelerating condition, while *Nlgn3ʰˢᵉ* mice exhibited a comparable performance to that of their wild-type littermates, suggesting that non-canonical NLGN3-PTPδ signaling is responsible for the enhanced motor learning.

**The autism-related *Nlgn3* R/C mutation disrupts the non-canonical NLGN3–PTPδ pathway.** The molecular pathogenic mechanism of the ASD-related R451C mutation remains unclear. Therefore, we examined the effect of the R451C mutation on the canonical and non-canonical synaptogenic pathways of NLGN3. Arg448 in mouse NLGN3, equivalent to Arg451 in human NLGN3, is located in the helix next to the dimer interface (Fig. 6a and Supplementary Fig. 4a). The side chain of Arg448 faces the protein core and forms a stacking interaction with Trp495. The R/C mutation seems to affect this local structure in close proximity to the dimer interface. In SEC purification, the R/C mutant of mouse NLGN3 was eluted as two peaks: the first, broad peak corresponds to higher-molecular-weight species larger than the wild-type dimer, whereas the second, sharp peak corresponds to the monomer (100 kDa; determined by SEC-MALS; Supplementary Fig. 4b). Structurally, the R/C mutation may affect the stability and dimeric assembly of NLGN3. Functionally, the R/C mutation reduced the pulldown efficiency by NRXN1β and PTPδA3B⁻ to 40% and 20% of wild-type, respectively (Fig. 6b). In the beads-neuron coculture assay, the staining signals of both Shank2 and gephyrin on the PTPδA3B⁻ beads were abolished in cortical neurons from *Nlgn3ᴿ/ᶜ* mutant mice (Fig. 6c). In contrast, the NRXN1β-induced accumulation of Shank2 and gephyrin was unaltered and significantly increased in *Nlgn3ᴿ/ᶜ* cortical neurons, respectively. These results suggest that the *Nlgn3* R/C mutation impairs the non-canonical pathway.

The resemblance of the synaptogenic properties of *Nlgn3ᴿ/ᶜ* and *Nlgn3ᵐᶠ* cortical neurons led us to further explore whether *Nlgn3ʰˢᵉ* and *Nlgn3ᵐᶠ* mutant mice exhibit the ASD-related endophenotypes observed in the *Nlgn3ᴿ/ᶜ* mice, such as selective increase in the expression of inhibitory synaptic proteins in the forebrain and GABA_A receptor-mediated inhibitory synaptic

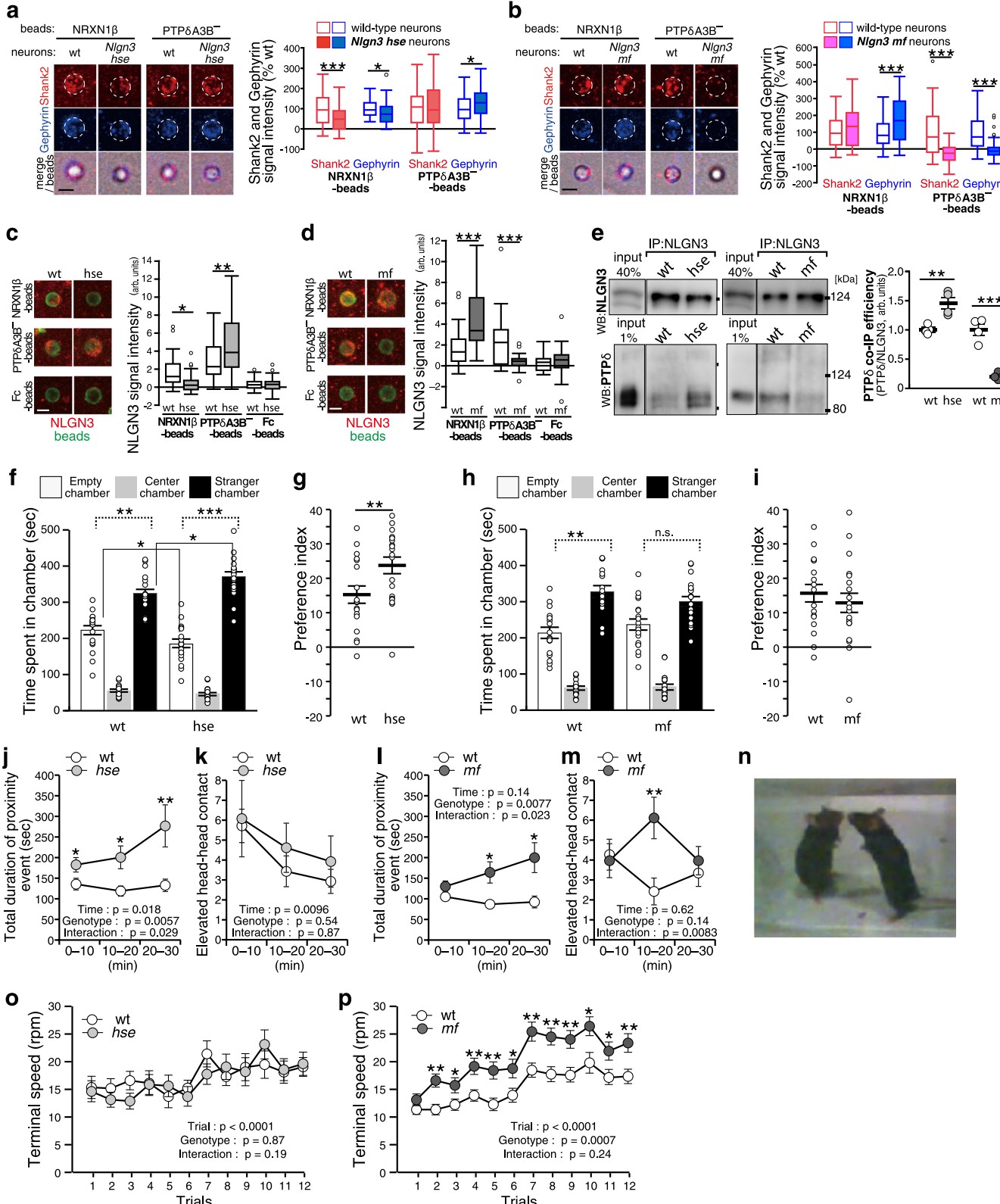

transmission in the cerebral cortical layer 2/3 neurons[15]. In *Nlgn3hse* mutant mice, no obvious changes in the expression levels of synaptic proteins in the forebrain and synaptic transmissions in the cerebral cortical layer 2/3 neurons were observed (Fig. 6d, e). In contrast, in the forebrain of *Nlgn3mf* mutant mice, the expression of inhibitory synaptic proteins vesicular GABA transporter (VGAT) and gephyrin, was

selectively upregulated, while those of the excitatory synaptic proteins vesicular glutamate transporter 1 (VGluT1) and PSD95 remained unaltered (Fig. 6d). Consistently, immunostaining signal ratios for VGAT compared to VGluT1 in the cerebral cortex were higher in the *Nlgn3mf* mutants than littermate controls (Supplementary Fig. 7j, k). On the contrary, the frequencies of both miniature excitatory and inhibitory

**Fig. 5 Differential regulation of social behavior by the canonical and non-canonical NLGN3 signaling. a–d** NRXN1β- and PTPδA3B⁻-induced postsynaptic differentiation (**a** and **b**) and NLGN3 accumulation (**c** and **d**) of cortical neurons from *Nlgn3ʰˢᵉ* (**a** and **c**) and *Nlgn3ᵐᶠ* (**b** and **d**) mice. Shank2, gephyrin, and NLGN3 immunostaining signals were quantified on the right ($n = 54$ and 66 NRXN1β-beads and 51 and 77 PTPδA3B⁻-beads for wild-type and *Nlgn3ʰˢᵉ* neurons, respectively for (**a**). $n = 51$ and 43 NRXN1β-beads and 57 and 60 PTPδA3B⁻-beads for wild-type and *Nlgn3ᵐᶠ* neurons, respectively for (**b**). $n = 35$ and 33 NRXN1β-beads, 32 and 36 PTPδA3B⁻-beads, and 30 and 34 Fc-beads for wild-type and *Nlgn3ʰˢᵉ* neurons, respectively for (**c**). $n = 28$ and 23 NRXN1β-beads, 29 and 34 PTPδA3B⁻-beads, and 22 and 26 Fc-beads for wild-type and *Nlgn3ᵐᶠ* neurons, respectively for (**d**). Scale bars, 5 μm. **e** Coimmunoprecipitation of NLGN3-PTPδ complex from striatal synaptosomal fraction of *Nlgn3ʰˢᵉ* and *Nlgn3ᵐᶠ* mutant mice. Representative images (left) and co-IP efficiency of PTPδ normalized to NLGN3 for immunoprecipitated samples (right) are shown ($n = 4$ experiments each for mutants and controls). **f–i** Three-chamber sociability test of *Nlgn3ʰˢᵉ* mutant mice (**f** and **g**) ($N = 19$ and 18 for mutants and controls, respectively) and *Nlgn3ᵐᶠ* mutant mice (**h** and **i**) ($N = 19$ and 17 for mutants and controls, respectively). Time spent in each chamber (**f** and **h**) and preference index (((time spent around stranger cage/ (Time spent around stranger cage + Time spent around empty cage)) × 100) – 50) (**g** and **i**) are presented. **j–n** Reciprocal social interaction test of *Nlgn3ʰˢᵉ* mice (**j** and **k**) ($N = 13$ and 14 pairs for mutant vs. mutant and control vs. control, respectively) and *Nlgn3ᵐᶠ* mice (**l** and **m**) ($N = 14$ and 13 pairs for mutant vs. mutant and control vs. control, respectively). Total duration of proximity events (**j** and **l**) and numbers of elevated head-head contact (**k** and **m**) during 30 min test are presented. **n** Elevated head-head contact observed in *Nlgn3ᵐᶠ* mice. **o** and **p** Rotation speed to fall off in each trial of *Nlgn3ʰˢᵉ* mice ($N = 17$ and 16 for mutants and controls, respectively) (**o**) and *Nlgn3ᵐᶠ* mice ($N = 37$ and 35 for mutants and controls, respectively) (**p**) in accelerating rotarod. Data in (**a–d**) are presented as box plots. The horizontal line in each box shows the median, the box shows the IQR and the whiskers are 1.5 × IQR. Data in (**e–m**, **o**, and **p**) represent mean ± s.e.m. *$p < 0.05$, **$p < 0.01$ and ***$p < 0.001$, Two-sided Student's *t*-test in (**a**, **b**, **e**, **f**, **j**, **l**, **m** and **p**), Mann–Whitney *U*-test in (**g**), Tukey's test in (**c** and **d**), and two-sided Bonferroni post-test in (**f** and **h**) (comparisons with dotted line). See Supplementary Table 4 for additional statistics and exact *p* values.

postsynaptic currents (mEPSC and mIPSC) were increased in *Nlgn3ᵐᶠ* mutants (Fig. 6f). The biochemical and histochemical phenotypes of *Nlgn3ᵐᶠ* mice, but not of *Nlgn3ʰˢᵉ* mutants, seem similar to those of the *Nlgn3^R/C^* ASD model mice, though electrophysiological ones are likely different.

## Discussion

NRXNs and type IIA RPTPs are the two major presynaptic hub molecules for organizing and specifying synapses through interactions with their cognate postsynaptic ligands[5,6]. In this study, we identified an interaction between PTPδ and NLGN3, one of the best-characterized postsynaptic binding partners of NRXNs[39,40]. This non-canonical interaction between NLGN3 and PTPδ is exclusively dependent on the deletion of the meB peptide and prefers meA peptide insertions in the Ig-like domains of PTPδ (Fig. 2). Our structural analyses suggest that the deletion of the meB peptide confines the spacing between Ig2 and Ig3 of PTPδ for simultaneous interaction with NLGN3, whereas the meA3 peptide in PTPδ Ig2 contributes to direct hydrophobic interaction with the C-terminal tail of the NLGN3 ECD. In fact, among the PTPδ-meB(−)s, the meA3-containing one showed the strongest NLGN3-binding activity in the CSB assay (Fig. 2c). Conversely, the synaptogenic activity of the PTPδ-meB(−)s became stronger as the length of the meA peptide increased (Fig. 1b), despite the observation that all the PTPδ-meB(−)s required NLGN3 for their synaptogenic activities (Supplementary Fig. 3). Therefore, additional factors may exist that reconcile this discrepancy between NLGN3-binding activity and the synaptogenic activity of PTPδ-meB(−)s. The NRXN-binding interface of NLGN3 overlapped with the PTPδ-binding interface and the shared site was crucial for binding to both proteins (Fig. 4d–f). This steric hindrance prevents NLGN3 from simultaneous interaction with NRXNs and PTPδ-meB(−)s, which may provide a molecular basis for the mutual counterbalancing of the canonical and non-canonical NLGN3 transsynaptic signaling for the developmental regulation of sociality, as discussed below. Structural comparison between the NLGN3–PTPδA3B⁻ and NLGN1–MDGA1 complexes show an overlap of their binding interfaces (Supplementary Fig. 2g), and MDGA1 can weakly interfere with the binding between NLGN3 and PTPδA3B⁻, as suggested by the competitive cell surface-binding assay mentioned above (Supplementary Fig. 2f). MDGA1 might function as a negative regulator for the PTPδ-meB(−)s–NLGN3 pathway as well as for the NRXN–NLGN pathway.

The *Nlgn3ʰˢᵉ* mutants impairing the canonical NLGN3-NRXN signaling showed increased social preference, whereas impairment of non-canonical signaling in the *Nlgn3ᵐᶠ* mutants caused a decrease in social preference and rather negative social response as defensive upright postures (Fig. 5f–n and Supplementary Fig. 8), suggesting the canonical and non-canonical NLGN3 signaling pathways contribute to social development in a different direction. Given that PTPδ-meB(−)s and NRXNs exhibited comparable $K_D$ values and sterically competed with each other for binding to NLGN3, PTPδ-meB(−)-mediated synaptogenesis is expected to be rather enhanced without the NLGN3–NRXN interaction in *Nlgn3ʰˢᵉ* mutant mice. In fact, more NLGN3-PTPδ complex was formed in the synaptosomal fraction of the *Nlgn3ʰˢᵉ* mouse brain and PTPδA3B⁻-mediated inhibitory synaptogenesis was increased in cultured *Nlgn3ʰˢᵉ* neurons (Fig. 5a–e). Conversely, NRXN1-mediated inhibitory synaptogenesis was increased in the *Nlgnᵐᶠ* neurons, which lacked PTPδA3B⁻-mediated synaptogenesis. These observations support the idea that the canonical and non-canonical synaptogenic pathways of NLGN3 actually compete and counterbalance with each other at synapse-level in the developing brain. Therefore, it may be reasonable to assume that the competition between the canonical and non-canonical NLGN3 synaptogenic signaling shapes local circuits within some brain regions, which, in turn, contributes to bidirectional regulation of social development.

Cortical neurons from the humanized *Nlgn3^R/C^* ASD model mice showed similar synaptogenic properties to those from *Nlgn3ᵐᶠ* mice, i.e., impaired PTPδ-meB(−)-mediated synaptogenesis and enhanced NRXN1-mediated inhibitory synaptogenesis (Fig. 6c). Furthermore, *Nlgn3ᵐᶠ* mice partly recapitulated the biochemical and histochemical features observed in the *Nlgn3^R/C^* ASD model, such as increased expression of inhibitory synaptic proteins in the forebrain; however, the synaptic phenotype in the cerebral cortices seemed different between the *Nlgn3^R/C^* mice and *Nlgn3ᵐᶠ* mice, in which the enhancement of synaptic transmission was not restricted to inhibitory synapses (Fig. 6d, f). Therefore, the selective disruption of the non-canonical NLGN3 pathway and the concomitant upregulation of the canonical pathway may cause the ASD-related endophenotypes of *Nlgn3^R/C^* mice. The R/C mutation causes ~90% decrease of the amount of NLGN3 protein in the mouse brain, probably due to disruption of proper dimeric conformation (Supplementary Fig. 4b) and missorting and degradation of the mutated proteins[12,15]. The upregulation of NRXN1-mediated inhibitory synaptogenesis in

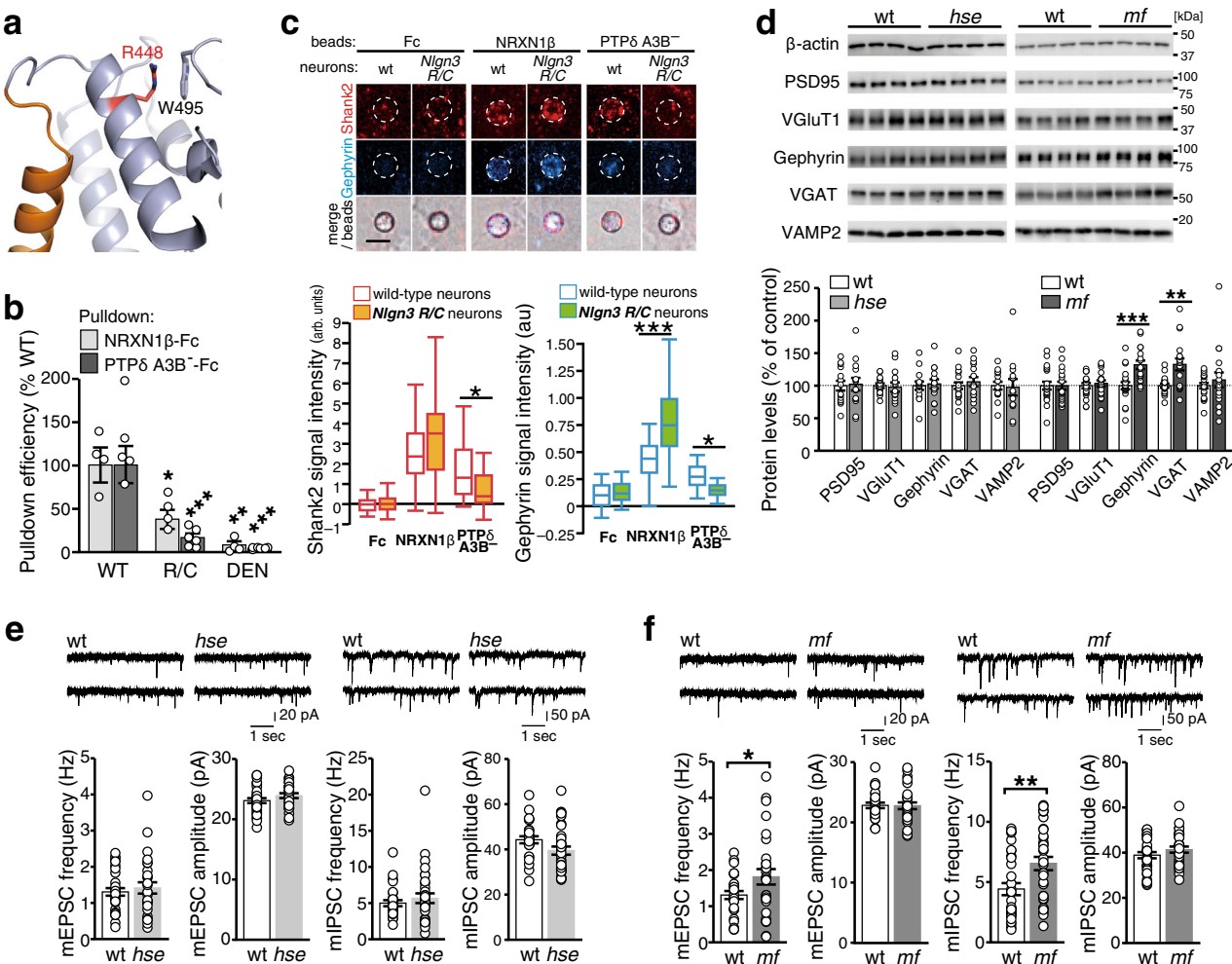

**Fig. 6 Phenotypic resemblance between *Nlgn3^R/C* ASD model and *Nlgn3^mf* mice. a** Close-up view of the area around Arg448 of NLGN3. Arg448 and Trp495 are shown as sticks. The coloring scheme is the same as that in Fig. 3a, except that Arg448 is colored in red. **b** Pulldown efficiencies of wild-type NLGN3 (WT), NLGN3-R448C (R/C), and NLGN3-DEN/AAA (DEN) by NRXN1β and PTPδA3B⁻ ($n = 4$ and 6 experiments for NRXN1β and PTPδA3B⁻, respectively). **c** NRXN1β- and PTPδA3B⁻-induced excitatory and inhibitory postsynaptic differentiation in *Nlgn3^R/C* cortical neurons were visualized by Shank2 (red) and gephyrin (blue) immunostaining (top) and quantified ($n = 27$ and 30 Fc-beads, 29 and 31 NRXN1β-beads, and 36 and 33 PTPδA3B⁻-beads for wild-type and *Nlgn3^R/C* neurons, respectively) (bottom). Scale bar, 5 μm. **d** Representative immunoblots (top) and quantification of expression levels (bottom) of synaptic proteins in *Nlgn3^hse* and *Nlgn3^mf* mutant mice and their wild-type littermates ($N = 14$ each for *Nlgn3^hse* and wild-type littermates; $N = 20$ and 18 for *Nlgn3^mf* and wild-type littermates, respectively). **e** and **f** Representative mEPSC and mIPSC traces and summary graphs from layer 2/3 pyramidal neurons of somatosensory cortex in acute slices of *Nlgn3^hse* (**e**) and *Nlgn3^mf* (**f**) mutant mice ($n = 28$, 31, 29, and 27 cells from 4 *Nlgn3^hse*, 6 wild-type, 5 *Nlgn3^mf*, and 5 wild-type mice, respectively for mEPSC; $n = 33$, 27, 32 and 31 cells from 6 *Nlgn3^hse*, 4 wild-type, 6 *Nlgn3^mf*, and 7 wild-type mice, respectively for mIPSC). Data in (**c**) are presented as box plots. Horizontal line in each box shows the median, box shows the IQR and the whiskers are 1.5 × IQR. Data in (**b**, **d**, **e**, and **f**) represent mean ± s.e.m. *$p < 0.05$, **$p < 0.01$ and ***$p < 0.001$, Tukey's test in (**b** and **c**), and two-sided Student's *t*-test in (**d** and **f**). See Supplementary Table 4 for additional statistics and exact *p* values.

*Nlgn3^R/C* neurons, despite of the decreased surface expression and NRXN1 binding affinity of the NLGN3 R451C protein, may suggest some ligand-level compensation mechanism involving other inhibitory postsynaptic NRXN1 ligands such as NLGN2 and Cblns. In contrast to the redundant interactions of NRXNs to all the NLGN family members, PTPδ-meB(−)s selectively interact with NLGN3 among NLGN family members and the synaptogenic activity of PTPδ-meB(−)s depends thoroughly on NLGN3 (Supplementary Fig. 3). Therefore, the substantial reduction of NLGN3 protein by the R/C mutation may selectively impair the non-canonical NLGN3 pathway with no functional compensation by other NLGNs. The robust increases of inhibitory synaptic proteins in the forebrain of *Nlgn3^R/C* or *Nlgn3^mf* mutants, despite of proportionally very low contribution of the non-canonical synaptogenic pathway, imply some circuit or system-level

mechanisms to compensate for the lack of the non-canonical NLGN3 pathway in these mice. The non-canonical versus canonical NLGN3 signaling model for social development could account for the phenotypic differences between *Nlgn3^R/C* and *Nlgn3* KO mice[15], in which both NLGN3 signaling pathways are expected to be abolished. The social phenotype of the *Nlgn3^R/C* model seems to depend largely on the genetic background, as social deficits are observed on a mixed 129Sv/C57BL/6 background, but not on a pure C57BL/6 background[15,41–43], which may relate to the non-robust social defect in *Nlgn3^mf* mutants on a C57BL/6 background in this study. In contrast, enhanced learning of motor routines, another aspect of the ASD-related behavioral phenotype observed in the *Nlgn3* KO and *Nlgn3^R/C* mice, likely regardless of genetic background, was selectively recapitulated by disruption of the non-canonical pathway in

$Nlgn3^{mf}$ mice (Fig. 5o,p). Therefore, the balancing mechanism of the canonical and non-canonical NLGN3 signaling associated with social development and the non-canonical pathway-selective regulatory mechanism for repetitive behaviors would be potential therapeutic targets for ASD.

Alternative splicing of ultra-short exons, called microexons, is most abundantly regulated in neuronal cells and is thought to modify the function of interaction domains of proteins in a switch-like manner for neuronal differentiation and function[44]. Misregulation of neural-specific alternative splicing of microexons is observed in individuals with ASD[45]. In line with this, we found that the meB and meA peptides of PTPδ regulated excitatory/inhibitory (E/I) synaptic target selection and the extent of synaptogenic activities, respectively. Dysregulation of meA/B splicing would therefore result in the disturbance of neuronal wiring including an imbalance of E/I synaptogenesis and a change in the non-canonical NLGN3-mediated synapse organization for ASD pathogenesis. Since the amount of the *Ptprd* meB(−) mRNAs in the developing mouse brain is <5% of total *Ptprd* mRNA[20], only a small number of neurons may express *Ptprd* meB(−) variants. The identification and characterization of the subset of synapses expressing PTPδ-meB(−)s in certain brain regions of wild-type and our mutant mice will help to understand the roles of the canonical and non-canonical NLGN3 signaling in each synapse as well as circuit basis of social development and repetitive behaviors linked to ASD pathogenesis. *Nlgn3* KO and $Nlgn3^{R/C}$ genetic mutations in mice have been shown to differently affect synaptic transmission at the calyx of Held synapse[46] and to exert input-selective effects in the hippocampal CA1 pyramidal neurons[17], implying a multiple transsynaptic ligand system for NLGN3 to regulate synaptic transmission and specificity in these synapses. Therefore, the competition mechanism between NRXNs and PTPδ may underlie the NLGN3-mediated developmental and functional regulation of these synapses. The mPFC and nucleus accumbens (NAc) are reported to be implicated in social behavioral changes in the $Nlgn3^{R/C}$ and *Nlgn3* KO models[18,47] and $Nlgn3^{mf}$ mutants exhibited a change of VGAT and VGluT1 expression ratio in the mPFC (Supplementary Fig. 7k). Furthermore, impaired synaptic inhibition onto D1-dopamine receptor-expressing neurons in NAc is known to be responsible for the enhanced motor learning in both $Nlgn3^{R/C}$ and *Nlgn3* KO models[11]. Synaptic functional analyses of $Nlgn3^{mf}$ mice in these brain regions and neurons would help to clarify the causal linkage between the lack of the non-canonical pathway and behavioral changes in sociability and motor learning. Recent structural studies have revealed that pre- and postsynaptic organizers form complexes in various combinations in a mutually exclusive manner by steric clashes[22–25,48]. Therefore, multiple synapse organizer pathways interact with each other, providing a molecular basis for the resilient and plastic development of neural networks.

## Methods

**Animal experiments.** Animal care and experimental protocols were approved by the Animal Experiment Committee of the University of Toyama (Authorization No. A2016MED-58 and A2016OPR-3) and conducted in accordance with the Guidelines for the Care and Use of Laboratory Animals of the University of Toyama. Mice were housed in a room with a 12-h light/dark cycle (lights on at 7:00 a.m.) at 23 ± 3 °C and 30–60% humidity with access to food and water ad libitum. Wild-type and mutant mice were generated by crossing wild-type male mice and heterozygous female mice. All biochemical, immunohistochemical, electrophysiological, and behavioral analyses were carried out with male mice. For behavioral testing, the male offspring of mating pairs were weaned around one month, genotyped, and housed 4 (two pairs of wild-type and mutant mice) per cage.

**Construction of expression vectors.** The coding sequences of mouse NLGN2, NLGN3, and NLGN4 were amplified by RT-PCR using total RNA prepared from the forebrains of mice at postnatal day (P) 11 with primer sets, Nlgn2-Cla-S/ Nlgn2-Sal-A, Nlgn3-ERI-S2/Nlgn3-Sal-A, and Nlgn4-ERI-S/Nlgn4-RV-A, respectively, and cloned into pFLAG-CMV-1 (Sigma) to yield N-terminally FLAG-tagged expression vectors. The FLAG-NLGN1 expression vector was previously described[20]. The coding sequences of ECDs of mouse PTPδ and NLGN splice variants were cloned into pEB6-Igk-Fc[49] to yield expression vectors for Fc fusion proteins. Expression vectors for the mutated forms of NLGN3 were generated by PCR-based mutagenesis using the FLAG-, Fc- or hexahistidine (His$_6$)-tagged NLGN3A (see "Crystallization") as templates. The coding sequences of ECD of mouse MDGA1, myc epitope, and His$_6$ were cloned into pEBMulti-Neo (Wako chemicals) to yield expression vector for myc-His-tagged MDGA1-ECD. Other expression vectors were previously reported[20,21,50]. Primer sequences used in this study are presented in Supplementary Table 3.

**Cell cultures and coculture assay.** Primary cortical cultures were prepared from mice at embryonic day (E) 18 or postnatal day (P) 0 essentially as described previously[20]. Cortical cells were placed on coverslips coated with 30 μg/mL poly-L-lysine and 10 μg/mL mouse laminin at a density of $5 \times 10^5$ cells per well in a 24-well-dish for coculture assay. The cells were cultured in Neurobasal-A supplemented with 2% B-27 supplement (Invitrogen), 5% fetal calf serum (FCS), 100 U/ mL penicillin, 100 μg/mL streptomycin and 0.5 mM L-glutamine for 24 h, and then cultured in the same medium without FCS. Cultures of HEK293T cells were maintained as previously described[51]. Expression vectors for Fc fusion proteins were transfected into HEK293T cells using NanoJuice transfection reagent (Novagen). After 5 days of culture, the culture media containing 1–30 μg/mL Fc fusion proteins were filtrated and incubated with Protein A-conjugated magnetic particles (smooth surface, 4.0–4.5 μm diameter; Spherotech). Beads coupled with Fc or Fc fusion proteins were added to cortical neurons at days in vitro (DIV) 13 to induce postsynaptic differentiation. After 24 h, cocultures were fixed and immunostained with rabbit anti-Shank2 (Frontier Institute, Shank2-Rb-Af750, 1:200), mouse anti-gephyrin (Synaptic Systems, 47011, 1:1000) or mouse anti-NLGN3 (Biolegend, MMS-5166-100, 1:200) antibodies. At least two independent neuronal cultures for each experimental condition were used for coculture experiments. Quantification of fluorescent signals of cocultures from *Nlgn3* mutant mice was conducted in a blind manner with respect to genotype.

**Screening of proteins interacting with PTPδA3B⁻.** Primary cortical cultures were prepared from ICR mice at P0 as previously described[20] and were placed on a cell culture dish coated with poly-L-lysine and laminin at a density of $1.5 \times 10^7$ cells per 100-mm dish. Magnetic beads coated with Fc or the ECD of PTPδA3B⁻ fused to Fc were added to cultured cortical neurons at DIV13 at a density of $5 \times 10^7$ beads per 100-mm dish. After 24 h, cultures were cross-linked with DTSSP (Pierce) and lysed according to previous protocols[26]. Bead-bound proteins were washed and separated by sodium dodecyl sulfate-polyacrylamide gel electrophoresis (SDS-PAGE). The gel lanes were excised into two pieces, one including the bait proteins and the other, reduced, alkylated, and digested with trypsin. The resulting peptides were analyzed by LC-MS/MS using an ESI ion trap mass spectrometer (LCQ Fleet, Thermo Fisher Scientific).

**CSB assay.** Expression vectors for FLAG-tagged NLGNs, PTPδA3B⁻ and NRXN1β were transfected into HEK293T cells. Transfected cells were incubated with Fc fusion recombinant proteins or control Fc (0.1 μM) in DMEM containing 10% FCS, 2 mM CaCl$_2$ and 1 mM MgCl$_2$ for 30 min at room temperature. After washing, cells were fixed with 4% PFA, immunostained with mouse anti-FLAG (Sigma) and rabbit anti-human IgG (Rockland) antibodies, and then visualized with Alexa Fluor 488-conjugated donkey anti-mouse IgG and Alexa Fluor 555-conjugated donkey anti-rabbit IgG antibodies (Invitrogen). For the competition assay, alkaline phosphatase tagged with Myc and His$_6$ epitopes (AP-MH), ECDs of MDGA1, NRXN1β and PTPδA3B⁻ tagged with Myc and His$_6$ epitopes (MDGA1-MH, NRXN1β-MH and PTPδA3B⁻-MH, respectively), and PTPδA3B⁻-Fc were added at concentrations of 0.01, 0.03, 0.1, 0.3, 1.0, and 3.0 μM. At least two independent experiments were performed for each experimental condition.

**Pulldown assay.** Soluble recombinant proteins were mixed at a concentration of 0.03 μM each in Hank's balanced salt solution (HBSS) containing 2 mM CaCl$_2$ and 1 mM MgCl$_2$ and incubated for 3 h followed by incubation with Protein A-Sepharose Fast Flow (GE Healthcare) for 1 h at room temperature. Subsequently, the sepharose suspensions were washed five times with HBSS containing 2 mM CaCl$_2$ and 1 mM MgCl$_2$. Bound proteins were eluted by boiling in SDS sample buffer, separated by a 5–20% gradient SDS-PAGE gel, and analyzed by Western blotting with horseradish peroxidase-conjugated mouse anti-His (MBL, M089-3, 1:1000).

**Cell aggregation assay.** HEK293T cells transfected with expression vectors for EGFP and PTPδ splice variants and those with expression vectors for RFP and NLGN3 were mixed at a density of $4.0 \times 10^6$ cells/mL and incubated at room temperature with rotation. After 15 min, fluorescence images were taken using a fluorescence stereomicroscope (M165 FC, Leica Microsystems). At least two independent experiments were conducted.

**Crystallization**. The gene encoding mouse NLGN3 ECD (residues 1–684) with the C-terminal His$_6$ tag was amplified by PCR from cDNA (accession No. NM_172932.4) with primers, Xho-Nlgn3-S and Not-Nlgn3-2052 (Supplementary Table 3), and cloned into the pEBMulti-Neo vector. The genes encoding the various lengths of mouse PTPδA3B⁻ ECD were cloned into the pEBMulti-Neo vector as described previously[24]. All proteins were transiently expressed using FreeStyle 293-F cells (Invitrogen). The proteins were purified from culture media by Ni-NTA (Qiagen) metal affinity resin following the standard protocol, and dialyzed against a buffer containing 20 mM Tris-HCl buffer (pH 8.0) containing 150 mM NaCl. The purified proteins were concentrated to ~100 μM, flash frozen in liquid N$_2$, and stored at −80 °C until use. For crystallization of apo-NLGN3 ECD, the protein solution was mixed at a 1:1 ratio with reservoir solution containing 15% PEG3350, 0.2 M ammonium chloride, and 0.1 M MES-Na (pH 6.0). The crystals of apo-NLGN3 ECD were grown at 20 °C by the sitting-drop vapor diffusion method. The crystals were cryoprotected by transferring crystals into a reservoir solution supplemented with 30% ethylene glycol. For crystallization of the NLGN3 ECD–PTPδ Ig1–Fn1 complex, each protein was mixed at an equimolar ratio. This mixture was further mixed at a 1:1 ratio with a reservoir solution containing 10% PEG3350, 0.1 M ammonium iodide, and 0.1 M MES-Na (pH 6.0). The crystals of the NLGN3–PTPδ Ig1–Fn1 complex were grown at 20 °C by the sitting-drop vapor diffusion method. The crystals were cryoprotected by transferring crystals into reservoir solution supplemented with 25% ethylene glycol and flash frozen in liquid N$_2$.

**Crystallography**. All diffraction data were collected at 100 K at BL41XU in SPring-8 and processed with the HKL2000[52] and CCP4 program suite[53] (Supplementary Table 2). The apo-NLGN3 ECD structure was determined by the molecular replacement method using the NLGN1 ECD (PDB 3BIX) as the search model with the program Molrep[54]. The NLGN3 ECD–PTPδ Ig1–Fn1 complex structure was determined by the molecular replacement method using the apo-NLGN3 ECD, PTPδ Ig1–Ig2 (PDB 2YD6)[55], PTPδ Ig3 (PDB 4YFD)[24] and PTPδ Fn1 (PDB 4YFE)[24] as the search models. Model building and refinement were carried out using the programs Coot[56] and Phenix[57], respectively. The final model of apo-NLGN3 ECD was refined at 2.76 Å to $R_{work}$ and $R_{free}$ of 0.218 and 0.251, respectively. The final model of the NLGN3 ECD–PTPδ Ig1–Fn1 complex was refined at 3.85 Å to $R_{work}$ and $R_{free}$ of 0.253 and 0.289, respectively. Even in this moderate resolution, positional refinement could be applied without secondary structure restraint or reference structure restraint for the final model. During the initial iterative cycles of model improvement and refinement, reference structure restraint was used to improve the main chain geometry. Buried surface areas were calculated by the program PISA[58]. The stereochemistry of the final models was assessed by the program MolProbity[59]. No residues are in the disallowed region for both structures. Structural figures were prepared with the program PyMol (Schrödinger, LLC).

**ITC analysis**. The NLGN3 ECD (wild-type or mutant) and PTPδ Ig1–Fn1 (A3B⁻ or A9B⁺) were dialyzed against 20 mM Tris-HCl buffer (pH 8.0) containing 150 mM NaCl. This dialysis process was repeated at least three times. For the measurement of the interaction between NLGN3 and NRXN1β, both proteins were dialyzed against the same buffer in the presence of 1 mM CaCl$_2$. The ITC measurements were performed at 25 °C with MicroCal Auto-iTC (GE Healthcare). Either wild-type or mutant NLGN3 (20 μM) was placed in the cell. PTPδ or NRXN1β solution (200 μM each) was added to the cell in a series of injections. Binding constants were calculated by fitting the data with a one-site binding model using the software Origin (MicroCal). Titration was performed twice with similar kinetics for wild-type proteins but once for NLGN3 mutant proteins due to technical difficulty related to protein expression, purification, and concentration.

**SEC-MALS**. Purified NLGN3 ECD (wild-type or R/C mutant) was concentrated to 1.5 or 1.1 g/L, respectively, and applied onto an ENrich SEC 650 (10 × 300 mm) column (Bio-Rad) pre-equilibrated with 20 mM Tris-HCl buffer (pH 7.5) containing 150 mM NaCl. The MALS data were collected on a DAWN HELEOS 8+ detector (Wyatt Technology) with an RF-20A UV detector (Shimadzu) and analyzed by the program ASTRA (Wyatt Technology).

**Generation of neuron-specific NLGN3 KO mice**. Neuron-specific Nlgn3 exon 7-deleted mice were obtained by crossing heterozygous female XX$^{Nlgn3fR/C}$ mice[15] with nestin-Cre transgenic male mice (+/nestin-cre) (Jackson Laboratories). The autism-related Nlgn3$^{R/C}$ mice were generated by the same crossing. Immediately after genotyping by PCR using tail-tip DNA with primers for Nlgn3 (Nlgn3-GT-A, Nlgn3-GT-B, and Nlgn3-GT-C, Supplementary Table 3) and Cre (Cre-S and Cre-A, Supplementary Table 3) at P0, the mice were sacrificed for cerebral cortical cultures.

**Generation of Nlgn3$^{mf}$ and Nlgn3$^{hse}$ knock-in mice**. The injection of guide RNAs, Cas9 RNA, and ssODNs into C57BL/6N mouse zygotes was performed as previously described[60]. The combinations of guide RNA and ssODN were as follows: gRNA-E6T2 and E6-ssODN1, and gRNA-E8T2 and E8-ssODN for the HSE/AAA and MF/AA mutations, respectively (For nucleotide sequences, see Supplementary

Table 3). The tail or ear tip of the F0 mice were alkaline lysed and subjected to cleaved amplified polymorphic sequence analyses using the following primer sets and restriction enzymes: Nlgn3-E6-U2/Nlgn3-E6-L2 and PvuII, and Nlgn3-E8-U2/Nlgn3-E8-L2 and BspHI, for the detection of the HSE/AAA and MF/AA mutations, respectively. The F0 mosaic mice were crossed with wild-type C57BL/6N mice to generate F1 founder mice. Exon 6, exon 8, and intron 6 of the Nlgn3 gene of the F1 mice were sequenced to confirm the precise knock-in events. The F2 heterozygous female mice were mated with wild-type C57BL/6N male mice to generate male mice for phenotypic analyses.

**Western blotting**. The S1 fractions were prepared from 6-week-old male mice as previously described[61], and protein concentrations were quantified using the BCA protein assay kit (PIERCE, Rockford, USA). Equal amounts of protein were separated by SDS-PAGE, transferred to PVDF membranes, and probed with primary antibodies followed by horseradish-peroxidase-conjugated secondary antibodies. The blots were developed and imaged using a Luminescent Image Analyzer LAS-4000 mini (Fujifilm, Tokyo, Japan). Blots were stripped and reprobed with antibodies against β-actin to normalize for differences in loading. The relative levels of each protein were determined by densitometric analysis using serially diluted protein samples on the same blots. The primary antibodies used in this study were as follows: mouse anti-NLGN3 (BioLegend (MMS-5166-100), 1:200 dilution), mouse anti-Gephyrin (Synaptic Systems (147111), 1:5000 dilution), mouse anti-VAMP2 (Synaptic Systems (104211), 1:2000 dilution), rabbit anti-actin (Santa Cruz Biotechnology (sc-1616-R), 1:1000 dilution), rabbit anti-PSD95 (a gift from Dr. Watanabe, Hokkaido University, 1:600 dilution), goat anti-VGAT (a gift from Dr. Watanabe, Hokkaido University, 1:600 dilution) and rabbit anti-VGluT1 (custom-made antibody for amino acid residues 542–560 of rat VGluT1 (GATHSTVQPPRPPPPVRDY), Eurofins Genomics, 1:1000 dilution). For both Nlgn3$^{mf}$ and Nlgn3$^{hse}$, four independent experiments using two litters (eight litters in total) were performed and pooled for densitometric analysis. Densitometric analysis of western blots was performed using Image J 1.46 software[62] in a genotype-blind manner. Uncropped blots are presented in the Source Data file.

**Real-time PCR**. Quantitative real-time PCR was performed using cDNAs prepared from olfactory bulb, cerebral cortex, hippocampus, striatum, thalamus, medulla oblongata, and cerebellum of 2 month-old mice with primer sets, Ptprd-725-S/Ptprd-941-A and Gapdh-S/Gapdh-A for total Ptprd and Gapdh, respectively. For amplification of the PtprdA3B⁻, the cDNAs were first subjected to 28 cycles of PCR with primers Ptprd-460-S and Ptprd-1907-A, and treated with Alu I and Rsa I to digest meA6 and/or meB-including Ptprd splice variants. Secondary PCR was performed using primers, PtprdA3-S and Ptprd-B-A. All the quantitative real-time PCR reaction was carried out on MX3000P (Stratagene) with GeneAce SYBR qPCR kit (Nippongene) to quantify the relative expression levels using the comparative threshold cycle (Ct) method. Ct values of total Ptprd were normalized by those of Gapdh. The relative expression of PtprdA3B⁻ in each brain region was quantified by comparing Ct values for PtprdA3B⁻ and total Ptprd. For primer sequences, see Supplementary Table 3.

**Coimmunoprecipitation**. The crude synaptosomal (S2′) fractions were prepared from striatums of 2–3 male mice of 4-week old as previously described[61] and treated with 1 mM DTSSP in a buffer containing 0.32 M sucrose, 4 mM HEPES (pH7.4) and 3 mM CaCl$_2$ for 30 min followed by incubation with 50 mM Tris-HCl (pH7.5) for 20 min. Synaptic proteins were solubilized with radio-immunoprecipitation (RIPA) buffer (50 mM Tris-HCl, pH8.0, 150 mM NaCl, 0.1% SDS, 0.5% sodium deoxycholate, 1% Nonidet P-40, 3 mM CaCl$_2$) containing protease inhibitors (Complete EDTA-free; Roche). Soluble fractions containing 0.8 mg protein were incubated with 3 μg rabbit anti-NLGN3 (Frontier Institute, Nlgn3-Rb-Af1010) or control rabbit IgG (Rockland, 609-4103) for 8 h followed by 4-h incubation with ~50 μg protein-G sepharose. The sepharose was washed five times with RIPA buffer. The bound proteins were boiled in SDS sample buffer containing 0.1 M DTT for 15 min to cleave DTSSP and to elute from the sepharose. The eluates were separated by SDS-PAGE, and analyzed by Western blotting with mouse anti-NLGN3 (Biolegend, 1:400 dilution) and mouse anti-PTPRD (Abcam (ab233806), 1:500 dilution) antibodies. For each Nlgn3$^{mf}$ and Nlgn3$^{hse}$ line, four independent experiments using one litter (4 litters in total) were performed and pooled for densitometric analysis. Uncropped blots are presented in the Source Data file.

**Immunohistochemistry**. Under deep isoflurane anesthesia, mice were perfused transcardially with 4% paraformaldehyde (PFA) in 0.1 M sodium phosphate buffer. After post-fixation with 4% PFA for 2 h, 50 μm sections were prepared with microslicer (VT1000S; Leica). Sections were incubated with rabbit anti-VGluT1 (a gift from Dr. Watanabe, Hokkaido University, 1:600 dilution) and goat anti-VGAT (a gift from Dr. Watanabe, Hokkaido University, 1:600 dilution), followed by incubation with Alexa Fluor 555-conjugated and Alexa Fluor 647-conjugated secondary antibodies (Invitrogen). For each Nlgn3$^{mf}$ and Nlgn3$^{hse}$ line, two independent experiments using one litter (2 liters in total) were performed and pooled for densitometric analysis. Images of double immunofluorescence were taken with a confocal laser-scanning microscope (SP5II; Leica) in a genotype-blind manner.

**Slice preparations**. Mice (13-16-days old) were anesthetized with sevoflurane anesthesia. After decapitation, their brains were removed from the skulls and rapidly submerged in an ice cold, oxygenated (95% $O_2$–5% $CO_2$ gas mixture) cutting solution containing (in mM) 252 sucrose, 3 KCl, 1.24 $NaH_2PO_4$, 26 $NaHCO_3$, 0.5 $CaCl_2$, 6.3 $MgSO_4$, 0.2 ascorbic acid, and 25 glucose. Coronal slices (300-μm thick) were cut on a vibrating blade microtome (PRO 7; Dosaka EM, Kyoto, Japan) in ice-cold oxygenated cutting solution. The slices, containing layer 2/3 of the somatosensory cortex, were incubated in oxygenated standard artificial cerebrospinal fluid (ACSF) containing (in mM) 126 NaCl, 3 KCl, 1.25 $NaH_2PO_4$, 26 $NaHCO_3$, 2 $CaCl_2$, 2 $MgSO_4$, and 25 glucose at 34 °C for 30 min. Recordings were performed after following incubation at room temperature for at least 30 min.

**Electrophysiological recordings**. The slices were transferred to a recording chamber with oxygenated ACSF continuously perfused at a flow rate of 1 ml/min at room temperature. Whole-cell patch-clamp recording pipettes were pulled from 1.5-mm thin-wall glass capillaries on an electrode puller (PP-830; Narishige, Tokyo, Japan). The whole cell recording pipettes contained (in mM) 145 KCl, 5 NaCl, 10 HEPES, 0.1 $CaCl_2$, 10 EGTA, 4 Mg-ATP, and 0.3 Na-GTP, and 10 QX-314 with pH adjusted to 7.3 with KOH. The pipette resistance was 3–6 MΩ. Neurons and recording pipettes were visualized using an infrared-differential interference contrast microscope (BX50WI; Olympus, Tokyo, Japan) with a ×40 water immersion objective and charge-coupled device camera (C2741-79; Hamamatsu Photonics, Hamamatsu, Japan) and a real-time differential video microscopy processor (XL-20; Olympus). Pyramidal neurons in layer 2/3 of the somatosensory cortex were identified by soma size and the single apical dendrite. Whole-cell voltage-clamp recording of the pyramidal neurons was performed with a holding potential of –70 mV. Data were corrected and digitized with a sampling rate of 20 kHz using a patch-clamp amplifier and an A/D converter board (Axopatch 200B and Digidata 1200B; Axon Instruments, Union City, CA). Signals were filtered at 1 kHz, and recorded on a hard disk using data acquisition and analysis software (pCLAMP 8; Axon Instruments). An Ag/AgCl reference electrode was placed near the intermediate position between the inlet and outlet of the chamber. To isolate miniature synaptic events, tetrodotoxin (0.5 μM) was contained in the ACSF at all times. In addition, the ACSF contained picrotoxin (50 μM) and 2-chloroadenosine (2 μM) for measurements of excitatory postsynaptic currents (EPSCs), or 6-cyano-7-nitroquinoxaline-2,3-dione (CNQX, 20 μM) for measurements of inhibitory postsynaptic currents (IPSCs). Miniature postsynaptic currents with an amplitude of >15 pA were detected on recording lasting 5 min or longer. Series resistance compensation was performed as much as possible by the amplifier. Series resistance was measured by application of a voltage step pulse (15 ms duration) every 10 s. Recordings with a series resistance of >20 MΩ or a leak current of >200 pA were discarded. All recordings and data analyses were performed in a genotype-blind manner.

**Three-chamber sociability test**. The test for sociability was conducted as previously described[36]. The apparatus comprised a rectangular, three-chambered box and a lid containing an infrared video camera (Ohara & Co.)[63]. Each chamber was 20 × 40 × 22 cm and the dividing walls were made from clear Plexiglas, with small square openings (5 × 3 cm) allowing access into each chamber. One day before testing, the subject mice were individually placed in the middle chamber and allowed to freely explore the entire apparatus for 10 min. In the sociability test, an unfamiliar C57BL/6J male (stranger) that had no prior contact with the subject mouse was placed in one of the side chambers. The placement of the stranger in the left or right side chamber was systematically alternated between trials. The stranger mouse was enclosed in a small, circular wire cage that allowed nose contact between the bars, but prevented fighting. The cage was 11 cm high, with a bottom diameter of 9 cm and bars spaced 0.5 cm apart. An identical empty wire cage was placed in the opposite chamber. A weighted plastic cup was placed on top of each cage to prevent the mouse from climbing to the top. The subject mouse was first placed in the middle chamber and allowed to explore the three chambers for 10 min. The amount of time spent in each chamber and in each area within 5 cm of the wire cage, the number of entries into each chamber and each area within 5 cm of the wire cage, total distance traveled, and average speed were recorded using Image J 1.47a software (NIH). Cage sniffing behavior during each 10-min test period was manually scored from video images of 600 frames (1 s/frame) by an observer blind to genotype. Sniffing time was defined as each frame in which a nose of a subject touched the wire cage.

**Reciprocal social interaction test and 3D video-based analysis of the social behavior**. The 3D video of the behavior of pairs of freely interacting mice was recorded and analyzed using the 3DTracker system[37] (www.3dtracker.org). With the aid of the system, a 3D image of mice in a transparent acrylic recording chamber (21 × 21 × 40 (H) cm), was reconstructed by integrating images captured by four depth cameras (Realsense R200, Intel Corp.) surrounding the chamber. Then, 3D positions of body parts (the center of head, neck, trunk, and hip) of each mouse in each video frame were estimated by fitting skeleton models of the mouse to the 3D images semi-automatically. The 3D video-based analysis enables quick and objective analysis of complex social behavior during the

reciprocal social interaction. For the test, adult male mutant mice and their wild-type littermates were used. The male mice were housed two to three animals per cage with their male littermates (wild-type and/or mutant mice) after weaning. A pair of mice for the test was randomly selected under the following constraints: (1) the mice had the same genotype; (2) the mice had not been housed together; (3) the body weight difference between the mice was <10%. Prior to the test day, the subject mice were individually placed in the recording chamber and allowed to freely explore for 20 min. In the test, a pair of mice were put in the recording chamber simultaneously, and were allowed to freely interact for 30 min. Some of the mice were tested once again in a different day after changing the pair combination. In the data analysis, to check the amount of the general social interaction, proximity events (distance of trunks of mice <7 cm) were counted. Although no obvious fighting was observed during the test, a proximity event could occur with negative (aggressive) as well as positive (affiliative) interaction. To further analyze the quality of the interaction in the proximity, taking advantages of the 3D video analysis, we also counted the defensive upright posture[38], as measured by elevated head-head contacts, where both mice were standing (height of head – height of hip > 3 cm) and facing in proximity (the distance of heads of mice <3 cm). The behavioral parameters were compared with two-way repeated measured ANOVA among six conditions: two groups (control and mutant groups) × 3-time intervals (0–10, 10–20, 20–30 min after the onset of the test). Subsequent multiple post hoc comparisons were performed with simple main effect analyses.

**Rotarod test**. Rotarod test was conducted using an accelerating rotarod apparatus (UGO Basile, Comerio-Varese, Italy) according to a previous report[11]. Mice were placed on rotating drum and the rotating speed (rpm) to fall was recorded with 5 min cutoff. Rotarod testing consisted of 12 trials with 3 trials per day over the course of 4 days. The speed of the drum accelerated from 4 to 40 rpm over a 5 min period for the first 6 trials, then from 8 to 80 rpm over a 5 min period for the next 6 trials.

**Image acquisition, quantification, and statistics**. Image acquisition and quantification were performed as described previously[20,49]. Statistical significance was evaluated by the Mann–Whitney U-test, Student's t-test or one-way ANOVA followed by Tukey's or Dunnett's post hoc test using JMP Pro software (version 15.0.0, SAS Institute Inc.). All the statistical tests used in this study were two-sided. Statistical significance was assumed when $p < 0.05$. The exact $p$ values for statistical tests used in this study are provided in Supplementary Table 4.

**Reporting summary**. Further information on research design is available in the Nature Research Reporting Summary linked to this article.

## Data availability
The coordinates and structure factors of apo-NLGN3 ECD and the NLGN3 ECD–PTPδ Ig1–Fn1 complex have been deposited in the Protein Data Bank under the accession codes 7CEE and 7CEG, respectively. Other data, mouse strains, and custom-made rabbit anti-VGluT1 antibody generated in this study are available from the corresponding authors upon reasonable request. Source data are provided with this paper.

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

## Acknowledgements

We thank Mses. Yumie Koshidaka, Emi Matsuura, Kayoko Awamori, Shoko Kawakami, Mizuki Sendo, and Maina Demura for technical assistance, Prof. Masahiko Watanabe (Hokkaido University) for antibodies against PSD95, VGluT1, and VGAT, and Prof. Toshihide Tabata (University of Toyama) for helpful suggestions. We also thank the beam-line staff at BL41XU of SPring-8 (Hyogo, Japan) for technical help during data collection and Enago (www.enago.jp) for English language review. This work was supported in part by grants from JSPS/MEXT KAKENHI (JP25293057, JP19H04744, JP20H03350, JP20K21444 and JP16H06276 (AdAMS) to T.Y., JP16H04749 to A.Y., JP16H04676 to M.M., JP16H06534 to J.M., and JP24247014 to S.F.), JST PRESTO and the Takeda Science Foundation to T.Y. and JST CREST (JPMJCR12M5) to T.U. and S.F.

## Author contributions

T.Y., T.S., A.I., S.I.-O. T.U., and K.A. performed biochemical, immunocytochemical, and cell biological analyses. T.Sh., Y.F., and M.F. performed mass spectrometry analysis. H.I., A.I., and T.Y. generated gene-edited *Nlgn3*$^{hse}$ and *Nlgn3*$^{mf}$ mice. J.K. performed

electrophysiological experiments and data processing. A.I., S.N., J.M., and K.Tak. performed behavioral experiments and data analysis. A.Y., T.S., and A.M. performed sample preparation and crystallization. A.Y. and S.F. collected diffraction data, analyzed the collected data, and determined the structures. F.O., T.Sa., and K.M. performed ITC measurement. S.T., K.Tab. and S.O. helped to prepare and analyzing *Nlgn3*[R/C] and *Nlgn3* KO mice. T.Y., A.Y., J.K., J.M., and S.F. wrote the manuscript with editing by K.M., M.F., H.N., K.Tak., H.M., A.Y., M.M., S.F., and T.Y. T.Y. and S.F. designed and supervised the study.

## Competing interests
The authors declare no competing interests.

## Additional information

Tomoyuki Yoshida [1,2,3✉], Atsushi Yamagata[4], Ayako Imai[1], Juhyon Kim[5], Hironori Izumi[1], Shogo Nakashima[6], Tomoko Shiroshima[7], Asami Maeda[8], Shiho Iwasawa-Okamoto[1], Kenji Azechi[1], Fumina Osaka[9], Takashi Saitoh[9], Katsumi Maenaka [9,10], Takashi Shimada[11], Yuko Fukata [12], Masaki Fukata [12], Jumpei Matsumoto[2,6], Hisao Nishijo [2,6], Keizo Takao [2,13], Shinji Tanaka[14], Shigeo Okabe [14], Katsuhiko Tabuchi [3,15,16], Takeshi Uemura [16,17], Masayoshi Mishina[18], Hisashi Mori [1,2] & Shuya Fukai [19✉]

[1]Department of Molecular Neuroscience, Faculty of Medicine, University of Toyama, Toyama, Japan. [2]Research Center for Idling Brain Science, University of Toyama, Toyama, Japan. [3]JST PRESTO, Saitama, Japan. [4]RIKEN Center for Biosystems Dynamics Research, Kanagawa, Japan. [5]Division of Bio-Information Engineering, Faculty of Engineering, University of Toyama, Toyama, Japan. [6]Department of System Emotional Science, Faculty of Medicine, University of Toyama, Toyama, Japan. [7]Department of Anatomy, Kitasato University School of Medicine, Kanagawa, Japan. [8]Research Institute for Diseases of Old Age, Juntendo University Graduate School of Medicine, Tokyo, Japan. [9]Center for Research and Education on Drug Discovery, Faculty of Pharmaceutical Sciences, Hokkaido University, Sapporo, Japan. [10]Laboratory of Biomolecular Science, Faculty of Pharmaceutical Sciences, Hokkaido University, Sapporo, Japan. [11]SHIMADZU Bioscience Research Partnership, Innovation Center, Shimadzu Scientific Instruments, Bothell, WA, USA. [12]Division of Membrane Physiology, Department of Molecular and Cellular Physiology, National Institute for Physiological Sciences, National Institutes of Natural Sciences, Aichi, Japan. [13]Life Science Research Center, University of Toyama, Toyama, Japan. [14]Department of Cellular Neurobiology, Graduate School of Medicine, The University of Tokyo, Tokyo, Japan. [15]Department of Molecular and Cellular Physiology, Institute of Medicine, Academic Assembly, Shinshu University, Nagano, Japan. [16]Institute for Biomedical Sciences, Interdisciplinary Cluster for Cutting Edge Research, Shinshu University, Nagano, Japan. [17]Division of Gene Research, Research Center for Supports to Advanced Science, Shinshu University, Nagano, Japan. [18]Brain Science Laboratory, Research Organization of Science and Technology, Ritsumeikan University, Shiga, Japan. [19]Department of Chemistry, Graduate School of Science, Kyoto University, Kyoto, Japan.
✉email: toyoshid@med.u-toyama.ac.jp; fukai@kuchem.kyoto-u.ac.jp

