## [Peer Review File · Nature Communications]

Reviewers' Comments:

Reviewer #1:

Remarks to the Author:

This is a very nice paper that reports a very important and unexpected result. Briefly, the authors described a novel interaction between the presynaptic cell adhesion molecule PTPRD and neuroligin 3 (NL3), the best characterized postsynaptic ligand for NRXNs. This interaction is regulated by alternative splicing of PTPRD, and occurs only with PTPRD isoforms missing MeB and containing MeA. The authors provide compelling evidence to support the existence of this non-canonical interaction, including the crystal structure of the complex between the PTPRD ECD (lacking MeB) and the NL3 ECD; cell aggregation assays and surface binding assays in HEK293 cells; pull down experiments with soluble proteins; and Kd measurements using ITC. Based on this information, they described that PTPRD and NRXNs interact with the same interface of NL3, and that the two proteins compete for the binding of NL3. The authors were also able to identify point mutations that disrupt selectively the binding of NL3 to NRXNs (hse-aaa) or PTPRD (mf-aa) and to generate knock-in mice with the aim of dissecting the functional role of NL3 canonical vs non-canonical pathways.

I believe that many of the experiments reported here are brilliant, and that this paper undoubtedly should be accepted for publication and published - this paper should have been in Nature, as it is far more novel than the avalanche of oxytocin-social behavior studies that have come out. So off the bat, my recommendation is to accept this paper for publication without changes.

Having said this, however, I would like to note a few weaknesses in this paper. I don't think the authors should address these weaknesses, except by revising the text and layout, but having read the paper I thought I might as well present them.

1. The strength of the paper are Figures 1-4, the discovery of a truly novel interaction that is likely physiologically significant. To appreciate the data, I would strongly recommend moving some of the SOM data to the main paper, such as the ITC data, the cell aggregation studies, and the heterologous synapse formation assays.

2. The generation of new NL3 knockin mice is heroic, and the behavioral data are useful (and fashionable), but the glaring hole in this paper is the lack of any functional characterization of synapses in the knockin mice. NL3 KO and two different strains of NL3 knockin mice have been extensively characterized physiologically. It would have been extremely informative whether the current knockin mice have phenotypes in the same synapses. The minimum the authors could do is to discuss these phenotypes, such as those described from my lab for the cerebellum, calyx, striatum, mPFC and hippocampus. Again, I don't suggest new experiments - electrophysiology is difficult, few labs can still do it, and few reviewers can still assess it. But in the end, it is essential to learn what actually happens at a synapse when specific protein interactions are disrupted, and neural circuits will never be understood without knowledge of their constituent synapses.

In summary, this is an important paper that I would recommend for publication with minor changes.

Thomas C. Sudhof

Reviewer #2:

Remarks to the Author:

This manuscript identifies a novel interaction between Neuroligin 3 (NL3) and protein tyrosine phosphatase delta (PTP). Their data suggest that PTP binding to NL3 competes with Neurexin binding to NL3. They identified partial structure of NL3/PTP complex to posit a competitive binding

mechanism between PTP and Neurexin. This allowed them to create mutant forms of NL3 that are more prone to bind to neurexins vs. PTP and vice versa. They then created knockin mouse models with two of the more selective NL3 mutants to determine how altering the balance between NL3 binding to Neurexin vs. to PTP affects behavioral outcomes compared to NL3 KO mice. They conclude that these two binding mechanisms to NL3 “fine-tune development of sociality and an imbalance in these pathways predisposes to autism”. The findings are quite novel and of interest to a broad neuroscience audience. The paper will be influential in the field by demonstration of a novel binding partner in competition with other known partners for NL3.

Using the mutant mouse models with differential selectivity of NL3 binding to Neurexin or PTP, they identify differences in behavior of these mutants. While the behavioral findings are clearly different, the interpretation of the behavioral findings is somewhat problematic. In the 3-chamber sociability task, their NL3hse mutant mouse model shows a significant preference for the social vs. inanimate target, as does the WT control; they do also report an increase in time spent with the social target between WT and NL3hse mutant and a significant decrease in time spent with the inanimate target between WT and NL3hse mutant. This is also reflected in significant difference in calculated preference index. They correctly interpret this as an increase in sociability, though some behavioral scientists prefer to interpret this task as “sociable or not sociable”, a statistically problematic way to interpret such data. In the same exact task, NL3mf mice show no statistically significant difference between WT and NL3mf mutants in either time spent with the social target or time spent with the inanimate target. They correctly interpret this as no change in sociability. This is also reflected in their lack of statistical difference on the preference index calculated from the same data set. It is important to point out that again, in spite of no statistically significant difference between WT and NL3mf behavior in this task, in light of the fact that they observe statistically significant preference for the social target in WT and no statistically significant preference for the inanimate target, some behavioral scientists would prefer to interpret these data as decreased sociability in the NL3mf mice. This latter interpretation is again statistically problematic. So this reviewer agrees with the authors’ interpretation of their 3-chamber sociability task. NL3hse mutants have increased sociability, and NL3mf mutants do not.

On a completely different task of direct, reciprocal, social interaction in freely moving pairs of mice, the authors find another difference in behavior between NL3hse vs. WT compared to NL3mf vs. WT. In the NL3hse mice, they find an increase in social proximity/interaction with no change in reared, head-to-head contact. In the NL3mf mice, they also find an increase in social proximity/interaction, but they also observe an increase in reared, head-to-head contact during 1/3 of the 30-minute test. They put their findings in this task together as NL3hse having increased affiliative social interaction and NL3mf having increased “aggressive” or “aversive” interaction (the reared, head-to-head interaction being increased during 1/3 of the task). Based on the difference in “elevated head-to-head contact” between NL3hse and NL3mf mice, there does appear to be a difference in overall behavior in this task. Interpretation of this as affiliative in one case and aversive in the other is, in this reviewer’s opinion, somewhat problematic. In the time period in NL3mf mice where no difference in “elevated head-to-head contact” is identified (20-30 minutes period), the NL3mf mice still exhibit significantly increased interaction just as the NL3hse mice do; in fact, the social interaction in both mutant models increases to the highest level during the 20-30 minute period (Figure 5, i-l). This reviewer feels that while the two mouse lines do in fact appear to behave differently, the interpretation of increased “positive” social interaction in NL3hse mutants and increased “negative” interaction in NL3mf mutants is over-interpreted. This reviewer would prefer a more agnostic approach and conclude that the pattern of behavior on social tasks in NL3hse and NL3mf mutants simply differs.

With respect to accelerating rotarod behavior (Fig. 5n-q), NL3hse mice do not differ from WT performance on this task. NL3mf mice show a significant main effect of genotype, indicating a statistically significant difference in their behavior on the rotarod task. Typically, one would interpret this finding as increased coordination or increased performance on the rotarod due to the genetic difference between NL3mf and WT cohorts. The authors instead would prefer to interpret their genotype effect as “enhanced acquisition of repetitive motor routines”. It is not clear that the interpretation of increased coordination vs. increased motor learning is critical to the overall findings of the paper. That said, using their ANOVA statistical analysis, there was no interaction

between genotype and time, indicating no statistically significant difference in rate of motor learning enhancement due to genotype. Thus a more appropriate conclusion is that NL3mf mice exhibit increased coordination or overall performance on the rotarod, rather than increased rate of motor learning. The authors do perform a very different style of analysis of their data by calculating a "learning rate" with units of rpm/trial. It is not clear how they derived the raw numbers used for this learning rate. For example, did they simply take the slope of a line between trial #1 and trial #12? The slope of a line fitted for each individual mouse's data? Or some other method? It is this reviewer's opinion that this is not a standard way to look at motor learning, and it is not statistically appropriate compared to ANOVA interaction between genotype and trial. Again, while this may slightly alter their conclusions, it does not impact their argument that the two mutants perform qualitatively differently on the Rotarod task compared to their respective WT controls. One additional piece of information the authors might use would be a statistical comparison of rotarod performance between NL3mf mice and WT mice on trial #1 of the rotarod (as though the rest of the trials were not collected) using a t-test, though ANOVA posthoc tests to determine if there is a difference in performance on day 1 may be more statistically appropriate. The authors would like to conclude that the two binding "pathways" of NL3 to neuexin and to PTP are in competition and that the balance between these two pathways is needed to "fine-tune" social interaction and rotarod behavior. Although this makes for a nice story, it is a bit removed from the actual data and underlying mechanisms. One can perhaps make the point that balance between these two molecular binding pathways can fine-tune synaptic function, as evidenced by differences in excitatory and inhibitory synapse markers in neuronal culture assays. One can

1. The interpretation of NL3mf vs. NL3hse behavior on the paired, reciprocal social interaction task is problematic as noted in detail above and should be changed to reflect a more agnostic point of view as to the quality of the difference.
2. Might there be a way to interpret existing video of the 3-chamber sociability task to determine if the NL3mf mutant interactions with the social target were in any way qualitatively different than those of the NL3hse mutant?
3. Interpretation of NL3mf rotarod behavior is problematic as noted in detail above and should be changed to better reflect the ANOVA lack of interaction between genotype and trial.
4. The conclusion that NL3 binding pathways fine-tune behavior is over-stated. The NL3 binding pathways alter synaptic function, which in turn may or may not be directly responsible for the observed behavioral differences (could be brain regions tested in the paper or other brain regions, for example).
5. There does not appear to be any synaptic phenotype at all in the NL3hse mutant brain region tested. So any interpretation about synaptic phenotypes is extrapolated from culture. Is there any synaptic difference in the NL3hse mice brains that might explain or at least lead one to suspect a rationale for the behavioral findings?
6. The "similarity" in synaptic phenotype between NL3mf and NL3R451C is over-stated. Yes, there is increased mIPSC frequency in layer 2/3 of cortex, but there is also increased mEPSC frequency. So it is really a different phenotype altogether. One may perhaps state that the mIPSC increased frequency phenotype is recapitulated in the NL3mf mutant, but many other, unrelated mutants may also have that same finding through entire different mechanisms.
7. It is worth pointing out clearly that the phenotypes in cultured neurons do not match what was found in layer 2/3 of the cortex in each model. This should be discussed in the discussion and the authors should be clear that it is not at all certain which of the few synaptic changes they observed could be driving behavioral differences, much less certain is which brain region(s).
8. Given #6 above, the conclusion that NL3R451C mutant synaptic phenotype is due to decreased PTPdelta interactions or changes in non-canonical NL3 pathway is not correct given the data. There are several assumptions and leaps of faith to suggest this from their findings in this reviewer's opinion.
9. It is not clear to this reviewer that the authors have sufficiently "separated" the two binding pathways to draw conclusions about a particular signaling pathway, canonical or non-canonical, drives a specific behavioral or synaptic change.

10. Therapeutic potential of the findings are a bit over-stated.
11. Details of how the reciprocal social interaction with pairs of mice was done are needed. Did they pair WT mice with WT and mutant with mutant? Did they use C57 or other WT mice as the social pair to the WT/mutant experimental mice? How were the mutant and WT mice housed, together or were mutants raised separately from WT?
12. Details of how the mice were bred are needed. hetXhet matings, or separate matings of WT X WT and homo X homo?
13. The authors cite refs 7-10 but there are additional papers examining the NL3R451C mutants to cite. This is also true of the citations to different genetic backgrounds later in the paper. Later in the paper, for example, they state that the social phenotype of the NL3R451C model is dependent on genetic background. They leave out reference to at least one of the papers demonstrating no phenotype on C57 background (Jaramillo et al, Autism Research 2018).
14. The authors repeatedly state that NL3R451C and NL3 KO mice "commonly" display enhanced formation of repetitive motor routines. They should drop "commonly" from all of these sentences. There is only one paper demonstrating enhanced performance on accelerating rotarod. Also, they should consider saying "display enhanced ability to remain on the accelerating rotarod" rather than "enhanced formation of repetitive motor routines" because the repetitive motor routines interpretation of this finding is highly controversial and is a conclusion not firmly supported by data.
15. Page 7 first full paragraph, "NLGN3 also interacted with LAR and PTPd splice variants lacking the meB peptide, although interactions were much weaker than those with PTPd-meB(-)s." It seems to this reviewer that "lacking the meB" and "PTPd-meB(-)s" are the same, i.e. both are lacking meB. So I think one of these two needs to be changed in this sentence.

Reviewer #3:

Remarks to the Author:

Yoshida and colleagues report a novel interaction of PTP proteins with Neuroligin 3, with PTPdeltaB- lacking mini-exon B its most prominent binding partner in vitro. The authors show that PTPdelta is synaptogenic in vitro and that its splice site B determines whether this activity is dependent on Neuroligin 3. They use structural biological studies to determine that Neuroligin 3 binds Nrnx1 and PTPdeltaB- through overlapping sites on Neuroligin 3 and use this information to design Neuroligin 3 point mutants that abrogate either Nrnx1 or PTPdeltaB- binding. In vitro experiments with these mutants support this approach and show that neurons expressing a Neuroligin 3 mutant that is impaired in Nrnx1 binding are less able to respond to Nrnx1 beads by assembling excitatory and inhibitory postsynaptic sites. Neurons expressing the second Neuroligin 3 mutant that is impaired in PTPdeltaB- binding in turn are unable to respond to the synaptogenic activity of PTPdeltaB-. Mice carrying these Neuroligin 3 point mutants were then developed. Mice with the Neuroligin 3 mutant that blocks Nrnx1 binding exhibit modestly enhanced social interaction whereas blocking PTPdeltaB- binding appears to have no effect. Motor learning was influenced in an opposite manner. Mice expressing the Neuroligin 3 mutant that blocks PTPdeltaB- binding have modestly increased excitatory and inhibitory mini frequencies. Finally, analysis of a mutant mouse line carrying the autism-associated Neuroligin 3 R451C shows impaired binding to both Nrnx1 and PTPdeltaB- and opposite effects on inhibitory postsynaptic induction by Nrnx1 and PTPdeltaB-, with the former increased and the latter decreased in Neuroligin 3 R451C mutant neurons.

The authors propose a model where competition between Nrnx1 and PTPdeltaB- binding to Neuroligin 3 controls inhibitory postsynaptic differentiation that regulates social behavior in mice. While this control of Neuroligin 3 activity through competing partners is an interesting idea, concerns exist about the extent to which the data support this. More detailed points are provided below.

Major points

1. The authors emphasize Nrnx1 and PTPdeltaB- ligand competition as the mechanism through which Neuroligin 3 activity is controlled. The data supporting ligand competition are very limited, though, and the in vitro cell-surface binding assay needs to be supported by additional evidence. Can PTPdeltaB- reciprocally displace cell-surface bound Nrnx1? Competitive binding should be demonstrated in the Nlgn3hse and Nlgn3mf mice neurons. Importantly, the authors should demonstrate increased Neuroligin 3-Nrxn1 interaction in Nlgn3mf mice in order to conclude that a PTPdeltaB- dependent change in this interaction modulates synapse development.
2. How is excitatory and inhibitory synapse density altered in neurons from the Nlgn3hse and Nlgn3mf mouse lines? This information is required to interpret the data from the bead-based assays of synapse induction. Also, to what extent are excitatory and inhibitory synapse density altered in the Nlgn3hse and Nlgn3mf mouse lines? This should be analyzed in regions relevant for the behavioral tests presented here.
3. Do Neuroligin 3 and PTPdelta form a complex in vivo? This should be tested using synaptic fractions. Also, given that Ptpd meB(-) mRNA abundance is less than 5% of total Ptpd mRNA, it is unclear how impactful any competition with Nrnx1 for binding to Neuroligin 3 is.
4. The social interaction data in Figure 5e and 5g are not sufficiently compelling to conclude that the Nlgn3hse and Nlgn3mf mouse lines have different social preferences, or that a possible modest difference is biologically meaningful.

Minor points

5. The authors interpret the data in Fig. 5a-d to suggest that "the canonical and non-canonical pathways seem to counterbalance each other for inhibitory synapse formation". This seems in conflict with data from Fig. 1 showing that Nrnx1 induces even more Gephyrin accumulation compared to PTPdeltaB-.
6. In Fig.6d, only total protein levels for the synaptic markers were analyzed and not their synaptic membrane levels. Such synaptic expression analysis will be necessary to make correlations with the electrophysiological data. Also, data for the synaptic protein levels for Shank2 and Gephyrin in the hse and mf mice will need to be shown to interpret the synaptogenic in vitro assays.
7. Can the author discuss what partner the more prominent isoform of PTPdelta containing splice site B engages to induce synaptic sites? Their data show that it is not Neuroligin 3. Is it IL1RAPL1/IL-1RAcP?
8. With respect to the idea that Nrnx1 and PTPdeltaB- compete with each other for Neuroligin 3 binding and that this controls social behavior, do the authors have data whether they are they coexpressed in the relevant circuits?
9. Since the R451C mutation reduces Neuroligin 3 expression by ~90% and in turn lowers surface trafficking, and since this mutation likely has dominant negative effects, the analysis of its role in dissecting extracellular synaptic interactions of Nlgn3 needs justification.

Reviewer #4:

Remarks to the Author:

In this contribution, Yoshida et al. report the identification of protein tyrosine phosphatase δ (PTP δ) as a novel interactor for Neuroligin-3 (NLGN3). They define this interaction as a non-canonical pathway as opposed to the canonical neurexin-neuroligin (NRXN-NLGN) pathway. By introducing structure-guided mutations into NLGN3, they aim to achieve pathway selectivity,

suggesting that balanced signalling through both pathways is important for proper development of sociality.

The unbiased discovery of PTP δ using proteomic approaches is convincing and ties in with the previous discovery of new binding partners for NRXN (notably LRRTM and Cbln) and for NLGN (notably MDGA). It is very interesting to see that signalling pathways initiated by Type IIA receptor protein tyrosine phosphatases, an abundant group of presynaptic hub proteins which bind many postsynaptic partners of their own, converge onto the postsynaptic NLGN hub, in this case the autism related NLGN3 specifically.

The work has important implications, is novel and is overall well executed. I support its acceptance if a number of major and minor issues are first resolved. From my domain of expertise, I will mainly comment on the molecular and structural aspects, as well as the overall biology.

Major remarks;

1. The absence of any discussion or experimental inclusion of MDGA as a major regulator of the canonical NRXN-NLGN interaction is remarkable and worrying. Discovered in 2013 and structurally described in 2017 by a trio of papers in *Neuron*, the MDGA-NLGN interaction has direct implications for this work, as MDGA proteins also compete with NRXN for binding to NLGN, using the same interface now also used by PTP δ . In other words, it is nearly certainly the case that MDGA will also act as a negative regulator for the NLGN3-PTP δ interaction/pathway described in this work. However, this is completely omitted from this work although it would significantly strengthen it further. It should be addressed using competition cell-binding or co-culture assays, by including a structural comparison of binding interfaces, and by including MDGA in the intro/discussion.

2. Spatial and temporal distribution of the NLGN3-PTP δ complex. The cell-based binding assays and bead-cortical neuron co-culture assays convincingly show that PTP δ and NLGN3 have the capacity to bind each other in a splice-dependent manner. However, there is no proof in the form of e.g. a proximity ligation assay or microscopic co-localisation that these two proteins actually meet in vivo in contacting neurons, which should be addressed either in co-cultured neurons or slices. The jump from in vitro cell-based experiments to full-blown in vivo and behavioural experiments is very abrupt, with little consideration for the fundamental synaptic biology of this novel complex.

3. The NLGN3-PTP δ -A3B- complex shows an interesting symmetric W-shaped complex with positions of N-and C-termini compatible with a trans-synaptic binding event. The nominal resolution of the crystallographic data is 4.15Å, which is low. The data have seemingly been processed to a correlation coefficient between random half data sets (CC1/2) cutoff value of ~ 0.5 . I suggest the authors try to squeeze out more data by including weaker reflections beyond this cutoff and by performing paired refinements to see if inclusion of these higher resolution reflections improves density, which can aid interpretation at this resolution.

4. Currently, there is not a single snapshot of the quality of the electron density map in the paper. A 2FoFc density map supplementary figure for the view in Fig. 3a contoured at a meaningful σ value should illustrate the quality of the density and increase confidence in domain placement. Similarly, in Fig. 4 a-d, putative side chain interactions are shown, however it is unlikely that at a resolution of 4.15Å most of them are resolved, let alone that they can be unambiguously placed. Again, corresponding 2FoFc density map supplementary figures must corroborate the accuracy of the model building in these interface regions and the conclusions drawn from it.

5. Related to the above comments, the "Crystallography" subsection within the "Methods" section does not detail how the low-resolution refinement was approached. How was the model restrained/constrained? Has only rigid body refinement been performed, or have side chains been positionally refined (which I infer), etc.? A more detailed description of the strategy for the low-resolution data processing, refinement and model building is warranted.

6. ITC analysis between NLGN3 and PTP δ -A3B- interestingly shows an endothermic reaction compensated by large increase of total system entropy. This is in contrast to the NLGN-NRXN interaction which is strongly exothermic by virtue of an interface Ca²⁺ ion. I note that the buffer background signal in almost all ITC experiments is high, often in the order of ~ 0.10 $\mu\text{cal/sec}$ (for example in Fig. 2g) and often of the same magnitude as the actual protein-protein signal. This may partially obscure the real protein-protein thermodynamic signal. It may be due to their use of the buffer Tris that has a large enthalpy of ionization (HEPES is generally better). Due to the low affinities between NLGN3 and PTP δ and NRXN proteins, the resolution c -value of the ITC thermograms is low, which, combined with the high buffer signal, obscures the inferred differences between WT, MF/IS, and HSE/AAA mutants; I am not convinced by the stated 6-fold reduction in K_d for the MF/IS mutant, based on the shape of the thermogram. Equilibrium SPR is a better option here for more accurate K_d determination and this complementary approach will strengthen the conclusions, as well as provide binding kinetics. For all ITC measurements, the error on the determination of ΔH , ΔS and K_d should be stated, which is also put out by the Origin analysis software.

7. PTP δ -A3B- MF/IS and MF/AA mutants were designed to specifically block the NLGN-PTP δ -A3B- interaction. The authors go on to use the MF/AA mutant in vivo. However, in Fig. S5 the ITC data for the MF/AA mutant is not shown. I realise the behavior of the MF/IS and MF/AA mutants is similar in the pull-down assay (Fig. 4g), but since the MF/AA mutant is used in vivo the corresponding biophysical data should formally be shown. It is also of note that the MF/IS mutation does actually not fully block binding of PTP δ -A3B- to NLGN3 but indeed still shows an apparent 6-times lower, but measurable K_d of ~ 27 μM (but see also comment above). It will then be interesting to see whether the MF/AA mutant is the same or blocks the interaction to a larger extent, compatible with the absence of Shank2 and Gephyrin signals (Fig. 5d). In the absence of a full interaction block, it raises questions about any residual signalling activity that may influence the behavioral data.

Minor remarks;

8. The authors should comment on whether the low resolution of the complex dataset may be due to the presence of the PTP δ Fn1 domain, which is very solvent exposed in the structure. Fn1 does not seem to be involved in making any contacts with NLGN3 and the rationale for including it is not clear; the paper does not contain any domain-deletion experiments to first converge on the minimal set of domains necessary for the interaction, which is normal practice. Deletion of Fn1 from the crystallisation construct may lead to a more defined sub-complex and better diffracting crystals.

9. Page 14, Discussion; "In fact, among the PTP δ -meB(-)s, the meA3-containing one showed the highest affinity to NLGN3"; with this cell-surface binding assay in Fig. 2B, true affinity is not really measured. An appropriate biophysical assay such as SPR that directly measures protein-protein interactions will be more accurate and conclusive, and with good probability show a better correlation with the synaptogenic effects of meA's of varying length.

10. It is well established that NLGN3 forms heterodimers with NLGN1 at excitatory synapses (Poulopoulos et al., 2012). Thus, at these synapses, there is the potential that NLGN1/NLGN3 heterodimers integrate NRX and PTP δ signalling. In that scenario, NRX and PTP δ would act synergistically instead of antagonistically. This hypothesis remains untested and is absent from the author's model, but it should at least be discussed in the light of the experimental findings.

11. The interface analysis focuses on competition of PTP δ with NRXN for binding to NLGN. Of course, there is also another aspect; the potential competition of NLGN with IL1RAPL1/IL-1RACp for binding to PTP δ ; the latter interaction was structurally described by the authors in 2015. A

structural comparison of NLGN vs. IL1RAPL1 for binding to PTP δ should be included with appropriate discussion.

Re: NCOMMS-20-30680-T

We are grateful to the reviewers for their helpful comments and made every effort to improve the manuscript according to their comments. We hope that our edits and the responses we provide below address the issues and concerns the reviewers have noted.

Comments from Reviewer #1:

This is a very nice paper that reports a very important and unexpected result. Briefly, the authors described a novel interaction between the presynaptic cell adhesion molecule PTPRD and neuroligin 3 (NL3), the best characterized postsynaptic ligand for NRXNs. This interaction is regulated by alternative splicing of PTPRD, and occurs only with PTPRD isoforms missing MeB and containing MeA. The authors provide compelling evidence to support the existence of this non-canonical interaction, including the crystal structure of the complex between the PTPRD ECD (lacking MeB) and the NL3 ECD; cell aggregation assays and surface binding assays in HEK293 cells; pull down experiments with soluble proteins; and Kd measurements using ITC. Based on this information, they described that PTPRD and NRXNs interact with the same interface of NL3, and that the two proteins compete for the binding of NL3. The authors were also able to identify point mutations that disrupt selectively the binding of NL3 to NRXNs (hse-aaa) or PTPRD (mf-aa) and to generate knock-in mice with the aim of dissecting the functional role of NL3 canonical vs non-canonical pathways.

I believe that many of the experiments reported here are brilliant, and that this paper undoubtedly should be accepted for publication and published - this paper should have been in Nature, as it is far more novel than the avalanche of oxytocin-social behavior studies that have come out. So off the bat, my recommendation is to accept this paper for publication without changes.

Having said this, however, I would like to note a few weaknesses in this paper. I don't think the authors should address these weaknesses, except by revising the text and layout, but having read the paper I thought I might as well present them.

1. The strength of the paper are Figures 1-4, the discovery of a truly novel interaction that is likely physiologically significant. To appreciate the data, I would strongly recommend moving some of the SOM data to the main paper, such as the ITC data, the cell aggregation studies, and the heterologous synapse formation assays.

2. *The generation of new NL3 knockin mice is heroic, and the behavioral data are useful (and fashionable), but the glaring hole in this paper is the lack of any functional characterization of synapses in the knockin mice. NL3 KO and two different strains of NL3 knockin mice have been extensively characterized physiologically. It would have been extremely informative whether the current knockin mice have phenotypes in the same synapses. The minimum the authors could do is to discuss these phenotypes, such as those described from my lab for the cerebellum, calyx, striatum, mPFC and hippocampus. Again, I don't suggest new experiments - electrophysiology is difficult, few labs can still do it, and few reviewers can still assess it. But in the end, it is essential to learn what actually happens at a synapse when specific protein interactions are disrupted, and neural circuits will never be understood without knowledge of their constituent synapses.*

In summary, this is an important paper that I would recommend for publication with minor changes.

Thomas C. Sudhof

Responses to the comments from Reviewer #1

1. *The strength of the paper are Figures 1-4, the discovery of a truly novel interaction that is likely physiologically significant. To appreciate the data, I would strongly recommend moving some of the SOM data to the main paper, such as the ITC data, the cell aggregation studies, and the heterologous synapse formation assays.*

According to the suggestion, we moved some of the data for NLGN3-PTP δ binding (cell aggregation assay) from previous Supplementary Fig. 2a to main Fig. 2d and edited the text accordingly.

2. *The generation of new NL3 knockin mice is heroic, and the behavioral data are useful (and fashionable), but the glaring hole in this paper is the lack of any functional characterization of synapses in the knockin mice. NL3 KO and two different strains of NL3 knockin mice have been extensively characterized physiologically. It would have been extremely informative whether the current knockin mice have phenotypes in the same synapses. The minimum the authors could do is to discuss these phenotypes, such as those described from my lab for the cerebellum, calyx, striatum, mPFC and hippocampus. Again, I don't suggest new experiments - electrophysiology is difficult, few labs can still do it, and few reviewers can still assess it. But in the end, it is essential to learn what actually happens at a*

synapse when specific protein interactions are disrupted, and neural circuits will never be understood without knowledge of their constituent synapses.

We appreciate the comment and agree with the reviewer that functional characterization of synapses in our knock-in mice is essential to understand the role of the novel transsynaptic interaction that we found. It is however difficult for us to experimentally address the functions of the NLGN3-PTP δ interaction and its competition with NLGN3-NRXNs in a single synapse at present, we instead discussed possible roles of the NLGN3-PTP δ interaction in synaptic development and function citing several pioneering works on NLGN3 KO and R451C models.

Page 18, Lines 4–16, “... *Nlgn3* KO and *Nlgn3*^{R/C} genetic mutations in mice have been shown to differently affect synaptic transmission at the calyx of Held synapse⁴⁶ and to exert input-selective effects in the hippocampal CA1 pyramidal neurons¹⁷, implying a multiple transsynaptic ligand system for NLGN3 to regulate synaptic transmission and specificity in these synapses. Therefore, the competition mechanism between NRXNs and PTP δ may underlie the NLGN3-mediated developmental and functional regulation of these synapses. The mPFC and nucleus accumbens (NAc) are reported to be implicated in social behavioral changes in the *Nlgn3*^{R/C} and *Nlgn3* KO models^{18,47} and *Nlgn3*^{mf} mutants exhibited a change of VGAT and VGLUT1 expression ratio in the mPFC (Supplementary Fig. 7i). Furthermore, impaired synaptic inhibition onto D1-dopamine receptor-expressing neurons in NAc is known to be responsible for the enhanced motor learning in both *Nlgn3*^{R/C} and *Nlgn3* KO models¹¹. Synaptic functional analyses of *Nlgn3*^{mf} mice in these brain regions and neurons would help to clarify the causal linkage between the lack of the non-canonical pathway and behavioral changes in sociability and motor learning.”

Comments from Reviewer #2:

This manuscript identifies a novel interaction between Neuroligin 3 (NL3) and protein tyrosine phosphatase delta (PTP). Their data suggest that PTP binding to NL3 competes with Neurexin binding to NL3. They identified partial structure of NL3/PTP complex to posit a competitive binding mechanism between PTP and Neurexin. This allowed them to create mutant forms of NL3 that are more prone to bind to neurexins vs. PTP and vice versa. They then created knockin mouse models with two of the more selective NL3 mutants to determine how altering the balance between NL3 binding to Neurexin vs. to PTP affects behavioral outcomes compared to NL3 KO mice. They conclude that these two binding mechanisms to NL3 “fine-tune development of sociality and an imbalance in these

pathways predisposes to autism". The findings are quite novel and of interest to a broad neuroscience audience. The paper will be influential in the field by demonstration of a novel binding partner in competition with other known partners for NL3.

Using the mutant mouse models with differential selectivity of NL3 binding to Neurexin or PTP, they identify differences in behavior of these mutants. While the behavioral findings are clearly different, the interpretation of the behavioral findings is somewhat problematic. In the 3-chamber sociability task, their NL3hse mutant mouse model shows a significant preference for the social vs. inanimate target, as does the WT control; they do also report an increase in time spent with the social target between WT and NL3hse mutant and a significant decrease in time spent with the inanimate target between WT and NL3hse mutant. This is also reflected in significant difference in calculated preference index. They correctly interpret this as an increase in sociability, though some behavioral scientists prefer to interpret this task as "sociable or not sociable", a statistically problematic way to interpret such data. In the same exact task, NL3mf mice show no statistically significant difference between WT and NL3mf mutants in either time spent with the social target or time spent with the inanimate target. They correctly interpret this as no change in sociability. This is also reflected in their lack of statistical difference on the preference index calculated from the same data set. It is important to point out that again, in spite of no statistically significant difference between WT and NL3mf behavior in this task, in light of the fact that they observe statistically significant preference for the social target in WT and no statistically significant preference for the inanimate target, some behavioral scientists would prefer to interpret these data as decreased sociability in the NL3mf mice. This latter interpretation is again statistically problematic. So this reviewer agrees with the authors' interpretation of their 3-chamber sociability task. NL3hse mutants have increased sociability, and NL3mf mutants do not.

On a completely different task of direct, reciprocal, social interaction in freely moving pairs of mice, the authors find another difference in behavior between NL3hse vs. WT compared to NL3mf vs. WT. In the NL3hse mice, they find an increase in social proximity/interaction with no change in reared, head-to-head contact. In the NL3mf mice, they also find an increase in social proximity/interaction, but they also observe an increase in reared, head-to-head contact during 1/3 of the 30-minute test. They put their findings in this task together as NL3hse having increased affiliative social interaction and NL3mf having increased "aggressive" or "aversive" interaction (the reared, head-to-head interaction being increased during 1/3 of the task). Based on the difference in "elevated head-to-head contact" between NL3hse and NLmf mice, there does appear to be a difference in overall behavior in this task. Interpretation of this as affiliative in one case and aversive in the other is, in this reviewer's opinion, somewhat problematic. In the time period in NL3mf mice where no difference in "elevated head-to-head contact" is identified (20-30 minutes period), the NL3mf mice still exhibit significantly increased interaction just as the NL3hse mice do; in fact, the social interaction in both mutant models increases to the highest level during the 20-30 minute period (Figure 5, i-l). This reviewer feels that while the two mouse lines do in fact appear to behave differently, the interpretation of increased "positive" social interaction in NL3hse mutants and increased "negative" interaction in NL3mf mutants is over-interpreted. This reviewer would prefer a

more agnostic approach and conclude that the pattern of behavior on social tasks in NL3hse and NL3mf mutants simply differs.

With respect to accelerating rotarod behavior (Fig. 5n-q), NL3hse mice do not differ from WT performance on this task. NL3mf mice show a significant main effect of genotype, indicating a statistically significant difference in their behavior on the rotarod task. Typically, one would interpret this finding as increased coordination or increased performance on the rotarod due to the genetic difference between NL3mf and WT cohorts. The authors instead would prefer to interpret their genotype effect as “enhanced acquisition of repetitive motor routines”. It is not clear that the interpretation of increased coordination vs. increased motor learning is critical to the overall findings of the paper. That said, using their ANOVA statistical analysis, there was no interaction between genotype and time, indicating no statistically significant difference in rate of motor learning enhancement due to genotype. Thus a more appropriate conclusion is that NL3mf mice exhibit increased

coordination or overall performance on the rotarod, rather than increased rate of motor learning. The authors do perform a very different style of analysis of their data by calculating a “learning rate” with units of rpm/trial. It is not clear how they derived the raw numbers used for this learning rate. For example, did they simply take the slope of a line between trial #1 and trial #12? The slope of a line fitted for each individual mouse’s data? Or some other method? It is this reviewer’s opinion that this is not a standard way to look at motor learning, and it is not statistically appropriate compared to ANOVA interaction between genotype and trial. Again, while this may slightly alter their conclusions, it does not impact their argument that the two mutants perform qualitatively differently on the Rotarod task compared to their respective WT controls. One additional piece of information the authors might use would be a statistical comparison of rotarod performance

between NL3mf mice and WT mice on trial #1 of the rotarod (as though the rest of the trials were not collected) using a t-test, though ANOVA posthoc tests to determine if there is a difference in performance on day 1 may be more statistically appropriate.

The authors would like to conclude that the two binding “pathways” of NL3 to neurexin and to PTP are in competition and that the balance between these two pathways is needed to “fine-tune” social interaction and rotarod behavior. Although this makes for a nice story, it is a bit removed from the actual data and underlying mechanisms. One can perhaps make the point that balance between these two molecular binding pathways can fine-tune synaptic function, as evidenced by differences in excitatory and inhibitory synapse markers in neuronal culture assays. One can

1. The interpretation of NL3mf vs. NL3hse behavior on the paired, reciprocal social interaction task is problematic as noted in detail above and should be changed to reflect a more agnostic point of view as to the quality of the difference.
2. Might there be a way to interpret existing video of the 3-chamber sociability task to determine if the NL3mf mutant interactions with the social target were in any way qualitatively different than those of the NL3hse mutant?
3. Interpretation of NL3mf rotarod behavior is problematic as noted in detail above and should be changed to better reflect the ANOVA lack of interaction between genotype and trial.

4. *The conclusion that NL3 binding pathways fine-tune behavior is over-stated. The NL3 binding pathways alter synaptic function, which in turn may or may not be directly responsible for the observed behavioral differences (could be brain regions tested in the paper or other brain regions, for example).*
5. *There does not appear to be any synaptic phenotype at all in the NL3hse mutant brain region tested. So any interpretation about synaptic phenotypes is extrapolated from culture. Is there any synaptic difference in the NL3hse mice brains that might explain or at least lead one to suspect a rationale for the behavioral findings?*
6. *The “similarity” in synaptic phenotype between NL3mf and NL3R451C is over-stated. Yes, there is increased mIPSC frequency in layer 2/3 of cortex, but there is also increased mEPSC frequency. So it is really a different phenotype altogether. One may perhaps state that the mIPSC increased frequency phenotype is recapitulated in the NL3mf mutant, but many other, unrelated mutants may also have that same finding through entire different mechanisms.*
7. *It is worth pointing out clearly that the phenotypes in cultured neurons do not match what was found in layer 2/3 of the cortex in each model. This should be discussed in the discussion and the authors should be clear that it is not at all certain which of the few synaptic changes they observed could be driving behavioral differences, much less certain is which brain region(s).*
8. *Given #6 above, the conclusion that NL3R451C mutant synaptic phenotype is due to decreased PTPdelta interactions or changes in non-canonical NL3 pathway is not correct given the data. There are several assumptions and leaps of faith to suggest this from their findings in this reviewer’s opinion.*
9. *It is not clear to this reviewer that the authors have sufficiently “separated” the two binding pathways to draw conclusions about a particular signaling pathway, canonical or non-canonical, drives a specific behavioral or synaptic change.*
10. *Therapeutic potential of the findings are a bit over-stated.*
11. *Details of how the reciprocal social interaction with pairs of mice was done are needed. Did they pair WT mice with WT and mutant with mutant? Did they use C57 or other WT mice as the social pair to the WT/mutant experimental mice? How were the mutant and WT mice housed, together or were mutants raised separately from WT?*
12. *Details of how the mice were bred are needed. hetXhet matings, or separate matings of WT X WT and homo X homo?*
13. *The authors cite refs 7-10 but there are additional papers examining the NL3R451C mutants to cite. This is also true of the citations to different genetic backgrounds later in the paper. Later in the paper, for example, they state that the social phenotype of the NL3R451C model is dependent on genetic background. They leave out reference to at least one of the papers demonstrating no phenotype on C57 background (Jaramillo et al, Autism Research 2018).*
14. *The authors repeatedly state that NL3R451C and NL3 KO mice “commonly” display enhanced formation of repetitive motor routines. They should drop “commonly” from all of these sentences. There is only one paper demonstrating enhanced performance on accelerating rotarod. Also, they should consider saying “display enhanced ability to remain on the accelerating rotarod” rather than “enhanced formation of repetitive motor routines” because the repetitive motor routines interpretation of this finding is highly controversial and is a conclusion not firmly supported by data.*

15. Page 7 first full paragraph, “NLGN3 also interacted with LAR and PTPd splice variants lacking the meB peptide, although interactions were much weaker than those with PTPd-meB(-)s.” It seems to this reviewer that “lacking the meB” and “PTPd-meB(-)s” are the same, i.e. both are lacking meB. So I think one of these two needs to be changed in this sentence.

Responses to the comments from Reviewer #2

1. The interpretation of *NL3mf* vs. *NL3hse* behavior on the paired, reciprocal social interaction task is problematic as noted in detail above and should be changed to reflect a more agnostic point of view as to the quality of the difference.

According to the suggestion, we revised the manuscript to simply describe the results of the paired, reciprocal social interaction test in the Results section as follows:

Page 13, Lines 5–12, “..... In the test, both *Nlgn3^{hse}* and *Nlgn3^{mf}* mutant mice spent more time in proximity with the other mice than their wild-type littermates (Fig. 5j,l and Supplementary Fig. 8i,j). We then examined a ‘defensive upright posture’³⁷ by measuring elevated head-head contacts during the test. Both *Nlgn3^{hse}* and their littermate wild-type mice showed comparable numbers of elevated head-head contacts throughout the test period while *Nlgn3^{mf}* mice exhibited the contacts more frequently than their wild-type littermates in the middle of the test session (Fig. 5m,n). These results suggest both *Nlgn3^{hse}* and *Nlgn3^{mf}* mutants showed increased social interaction but their qualities seem different.”

2. Might there be a way to interpret existing video of the 3-chamber sociability task to determine if the *NL3mf* mutant interactions with the social target were in any way qualitatively different than those of the *NL3hse* mutant?

We appreciate the suggestion on analyses of video source during the 3-chamber sociability test. We analyzed sniffing behavior of our mutants during the test and found *Nlgn3^{mf}* mice showed significantly shorter duration of each sniffing to the stranger mouse compared to littermate wild-type mice. We added new figures (Supplementary Fig. 7g,h) and results on these analyses in the Results section as follows:

Page 12, Line 28 – Page 13, Line 3, “... Although the social-behavioral phenotype of *Nlgn3^{mf}* mice was not robust, *Nlgn3^{mf}* mice showed significantly shorter duration in each

stranger-cage-sniffing behavior than their wild-type littermates (Fig. 5h,i and Supplementary Fig. 7d–f,h”).

3. Interpretation of NL3^{mf} rotarod behavior is problematic as noted in detail above and should be changed to better reflect the ANOVA lack of interaction between genotype and trial.

We removed the figures of learning rates of our mutants (previous Fig. 5o,p) and revised the interpretation of rotarod behavior of *Nlgn3^{mf}* mice according to the reviewer’s suggestion as follows:

Page 13, Lines 16–19, “*Nlgn3^{mf}* mice showed a significantly higher performance than wild-type littermates in this accelerating condition, while *Nlgn3^{hse}* mice exhibited a comparable performance to that of their wild-type littermates, suggesting that non-canonical NLGN3-PTPδ signaling seems responsible for the enhanced motor learning.”

4. The conclusion that NL3 binding pathways fine-tune behavior is over-stated. The NL3 binding pathways alter synaptic function, which in turn may or may not be directly responsible for the observed behavioral differences (could be brain regions tested in the paper or other brain regions, for example).

We weakened the statement on the behavioral regulation by the competitive NLGN3 binding pathways in the Abstract by rephrasing “fine-tune” as follows. Also, we weakened statement on relationship between imbalance in these pathways and ASD according to the comment #10 below.

Abstract, the last sentence, “... Our findings suggest that canonical and non-canonical NLGN3 pathways compete to fine-tune the development of sociality and an imbalance in these pathways predisposes to autism.” >> “... Our findings suggest that canonical and non-canonical NLGN3 pathways compete and regulate the development of sociality and an imbalance in these pathways may predispose to autism.”

5. There does not appear to be any synaptic phenotype at all in the NL3^{hse} mutant brain region tested. So any interpretation about synaptic phenotypes is extrapolated from culture. Is there any synaptic difference in the NL3^{hse} mice brains that might explain or at least lead one to suspect a rationale for the

behavioral findings?

Any synaptic changes in excitatory/inhibitory synaptic transmission and protein expression levels that might account for behavioral phenotypes of the *Nlgn3^{hse}* mice have not yet been detected in our hands. The only change we could detect in the brain of the *Nlgn3^{hse}* line was an increase in the amount of NLGN3-PTP δ complex in the synaptosomal fraction (Fig. 5e), which is in line with our hypothesis that NLGN3-PTP δ pathway competes with NLGN3-NRXN pathway in vivo, though is still not directly linked to the behavioral changes. We, therefore, discussed about possible interpretations of mouse behavior from these in vivo data in the Discussion section as follows:

Page 16, Lines 4–6, “...In fact, more NLGN3-PTP δ complex was formed in the synaptosomal fraction of the *Nlgn3^{hse}* mouse brain and PTP δ A3B⁻-mediated inhibitory synaptogenesis was increased in cultured *Nlgn3^{hse}* neurons (Fig. 5a–e).”

Page 16, Lines 10–12, “...Therefore, it may be reasonable to assume that the competition between the canonical and non-canonical NLGN3 synaptogenic signaling shapes local circuits within some brain regions, which, in turn, contributes to bidirectional regulation of social development.”

6. The “similarity” in synaptic phenotype between *NL3mf* and *NL3R451C* is over-stated. Yes, there is increased mIPSC frequency in layer 2/3 of cortex, but there is also increased mEPSC frequency. So it is really a different phenotype altogether. One may perhaps state that the mIPSC increased frequency phenotype is recapitulated in the *NL3mf* mutant, but many other, unrelated mutants may also have that same finding through entire different mechanisms.

We agree that synaptic electrophysiological phenotypes between *Nlgn3^{mf}* and *NL3R451C* are different. In contrast, biochemical (Fig. 6d) and immunohistochemical (Supplementary Fig. 6i) as well as in vitro cultural phenotypes (Fig. 6c) are quite similar each other. Therefore, we revised the manuscript to separately and carefully describe these phenotypes and weakened the statement on the linkage between R451C phenotypes and the impaired non-canonical NLGN3 pathway in the Results and Discussion sections.

Page 14, Lines 19–25, “... Consistently, immunostaining signal ratios for VGAT compared to VGluT1 in the cerebral cortex were higher in the *Nlgn3^{mf}* mutants than littermate controls (Supplementary Fig. 6h,i). On the contrary, the frequencies of both miniature excitatory and

inhibitory postsynaptic currents (mEPSC and mIPSC) were increased in *Nlgn3^{mf}* mutants (Fig. 6f). The biochemical and histochemical phenotypes of *Nlgn3^{mf}* mice, but not of *Nlgn3^{hse}* mutants, seem similar to those of the *Nlgn3^{R/C}* ASD model mice, though electrophysiological ones are likely different.”

Page 16, Lines 20–21, “... Therefore, the selective disruption of the non-canonical NLGN3 pathway and the concomitant upregulation of the canonical pathway may cause the ASD-related endophenotypes of *Nlgn3^{R/C}* mice.”

7. It is worth pointing out clearly that the phenotypes in cultured neurons do not match what was found in layer 2/3 of the cortex in each model. This should be discussed in the discussion and the authors should be clear that it is not at all certain which of the few synaptic changes they observed could be driving behavioral differences, much less certain is which brain region(s).

We think it difficult to directly compare the data of cultured neurons i.e. synaptogenic properties and those of layer 2/3 electrophysiology because they are quite different level of data; the former is just to estimate amount of synapse-organizing signals induced by beads, but the latter to measure synaptic function. However, we absolutely agree with the reviewer that it is not at all clear how the synapse-organizer dysregulation leads to behavioral changes and which brain regions are responsible. Regarding this, we discussed future prospects of our study to link synaptic changes to behavioral phenotypes.

Page 17, Line 28–Page 18, Line 4, “The identification and characterization of the subset of synapses expressing PTP δ -meB(-)s in certain brain regions of wild-type and our mutant mice will help to understand the roles of the canonical and non-canonical NLGN3 signaling in each synapse as well as circuit basis of social development and repetitive behaviors linked to ASD pathogenesis.”

Page 18, Lines 9–16, “The mPFC and nucleus accumbens (NAc) are reported to be implicated in social behavioral changes in the *Nlgn3^{R/C}* and *Nlgn3* KO models^{18,47} and *Nlgn3^{mf}* mutants exhibited a change of VGAT and VGluT1 expression ratio in the mPFC (Supplementary Fig. 7i). Furthermore, impaired synaptic inhibition onto D1-dopamine receptor-expressing neurons in NAc is known to be responsible for the enhanced motor learning in both *Nlgn3^{R/C}* and *Nlgn3* KO models¹¹. Therefore, synaptic functional analyses of *Nlgn3^{mf}* mice in these brain regions would help to clarify the causal linkage between the lack of the non-canonical pathway and behavioral changes in sociability and motor learning.”

8. Given #6 above, the conclusion that NL3R451C mutant synaptic phenotype is due to decreased PTPdelta interactions or changes in non-canonical NL3 pathway is not correct given the data. There are several assumptions and leaps of faith to suggest this from their findings in this reviewer's opinion.

We weakened the description on the linkage between R451C phenotypes and the impaired non-canonical NLGN3 pathway. Please see the responses to the comment #6.

9. It is not clear to this reviewer that the authors have sufficiently "separated" the two binding pathways to draw conclusions about a particular signaling pathway, canonical or non-canonical, drives a specific behavioral or synaptic change.

Our binding assays, heterologous synapse formation assays, and newly added immunocytochemical analyses for NLGN3 (Fig. 5c–e) strongly suggest that the NLGN3 HSE/AAA and MF/AA mutations block interactions to and signaling pathways through NRXNs and PTP δ -meB(–)s in neurons, respectively. However, in mice carrying these mutations, it could not be excluded that these mutations may cause some compensatory mechanisms that contribute to synaptic and behavioral phenotypes. In fact, despite the amount of synaptogenesis mediated directly by the non-canonical NLGN3-PTP δ -meB(–)s pathway is expected to be very low, the MF/AA mutation in mice caused robust changes in expression levels of synaptic proteins in whole forebrain (Fig. 6d). So, we added discussion about the compensatory mechanisms.

Page 17, Lines 4–7, "The robust increases of inhibitory synaptic proteins in the forebrain of *Nlgn3*^{R/C} or *Nlgn3*^{mf} mutants, despite of proportionally very low contribution of the non-canonical synaptogenic pathway, imply some circuit or system-level mechanisms to compensate for the lack of the non-canonical NLGN3 pathway in these mice."

10. Therapeutic potential of the findings are a bit over-stated.

We weakened the statement on the therapeutic potential of our findings in Summary and Discussion sections. Please see the responses to the comment #4.

11. Details of how the reciprocal social interaction with pairs of mice was done are needed. Did they pair

WT mice with WT and mutant with mutant? Did they use C57 or other WT mice as the social pair to the WT/mutant experimental mice? How were the mutant and WT mice housed, together or were mutants raised separately from WT?

We only tested mutant vs. mutant and wild-type vs. wild-type pairs for the reciprocal social interaction. This is stated in the revised figure legend for Fig. 5j–n. According to the comment, we added the following text in the Methods (“**Reciprocal social interaction test and 3D video-based analysis of the social behavior**” section).

Page 29, Lines 6–10, “For the test, adult male mutant mice and their wild-type littermates were used. The male mice were housed two to three animal per cage with their male littermates (wild-type and/or mutant mice) after weaning. A pair of mice for the test was randomly selected under the following constraints: 1) the mice had a same genotype; 2) the mice had not been housed together; 3) the body weight difference between the mice was < 10%.”

12. Details of how the mice were bred are needed. hetXhet matings, or separate matings of WT X WT and homo X homo?

Heterozygous female mice (XX^{hse} or XX^{mf}) and wild-type (XY) male mice were mated to obtain mutant ($X^{hse}Y$ or $X^{mf}Y$) and wild-type (XY) male mice for behavioral, electrophysiological, and biochemical analyses. We added the details of how the mice were bred in the Methods (“**Animal experiments**” section).

Page 19, Lines 1–6, “Mice were housed in a room with a 12 h light/dark cycle (lights on at 7:00 a.m.) with access to food and water ad libitum. Wild-type and mutant mice were generated by crossing wild-type male mice and heterozygous female mice. All biochemical, immunohistochemical, electrophysiological, and behavioural analyses were carried out with male mice. For behavioural testing, the male offspring of mating pairs were weaned around one month, genotyped, and housed 4 (two pairs of wild-type and mutant mice) per cage.”

13. The authors cite refs 7-10 but there are additional papers examining the NL3R451C mutants to cite. This is also true of the citations to different genetic backgrounds later in the paper. Later in the paper, for example, they state that the social phenotype of the NL3R451C model is dependent on genetic

background. They leave out reference to at least one of the papers demonstrating no phenotype on C57 background (Jaramillo et al, Autism Research 2018).

We appreciate the reviewer for letting us know about a paper examining the R451C mutants. We cited Jaramillo et al, Autism Research 2018 in the Discussion section as ref #43.

14. The authors repeatedly state that NL3R451C and NL3 KO mice “commonly” display enhanced formation of repetitive motor routines. They should drop “commonly” from all of these sentences. There is only one paper demonstrating enhanced performance on accelerating rotarod. Also, they should consider saying “display enhanced ability to remain on the accelerating rotarod” rather than “enhanced formation of repetitive motor routines” because the repetitive motor routines interpretation of this finding is highly controversial and is a conclusion not firmly supported by data.

According to the suggestion, we removed the word “commonly” from all of the relevant sentences. We also adopted the sentence “display enhanced ability to remain on the accelerating rotarod” instead of “enhanced formation of repetitive motor routines.

15. Page 7 first full paragraph, “NLGN3 also interacted with LAR and PTPd splice variants lacking the meB peptide, although interactions were much weaker than those with PTPd-meB(-)s.” It seems to this reviewer that “lacking the meB” and “PTPd-meB(-)s” are the same, i.e. both are lacking meB. So I think one of these two needs to be changed in this sentence.

We agree that the original sentence was a little bit confusing, and revised the sentence as follows:

Page 7, Lines 12–14, “... NLGN3 also interacted with LAR and PTP σ splice variants lacking the meB peptide, although interactions were much weaker than the interaction with PTP δ A3B $^{-}$ (Supplementary Fig. 2a).”

Comments from Reviewer #3:

Yoshida and colleagues report a novel interaction of PTP proteins with Neuroligin 3, with PTPdeltaB-lacking mini-exon B its most prominent binding partner in vitro. The authors show that PTPdelta is synaptogenic in vitro and that its splice site B determines whether this activity is dependent on

Neurologin 3. They use structural biological studies to determine that Neurologin 3 binds Nrxn1 and PTPdeltaB- through overlapping sites on Neurologin 3 and use this information to design Neurologin 3 point mutants that abrogate either Nrxn1 or PTPdeltaB- binding. In vitro experiments with these mutants support this approach and show that neurons expressing a Neurologin 3 mutant that is impaired in Nrxn1 binding are less able to respond to Nrxn1 beads by assembling excitatory and inhibitory postsynaptic sites. Neurons expressing the second Neurologin 3 mutant that is impaired in PTPdeltaB- binding in turn are unable to respond to the synaptogenic activity of PTPdeltaB-. Mice carrying

these Neurologin 3 point mutants were then developed. Mice with the Neurologin 3 mutant that blocks Nrxn1 binding exhibit modestly enhanced social interaction whereas blocking PTPdeltaB- binding appears to have no effect. Motor learning was influenced in an opposite manner. Mice expressing the Neurologin 3 mutant that blocks PTPdeltaB- binding have modestly increased excitatory and inhibitory mini frequencies. Finally, analysis of a mutant mouse line carrying the autism-associated Neurologin 3 R451C shows impaired binding to both Nrxn1 and PTPdeltaB- and opposite effects on inhibitory postsynaptic induction by Nrxn1 and PTPdeltaB-, with the former increased and the latter decreased in Neurologin 3 R451C mutant neurons.

The authors propose a model where competition between Nrxn1 and PTPdeltaB- binding to Neurologin 3 controls inhibitory postsynaptic differentiation that regulates social behavior in mice. While this control of Neurologin 3 activity through competing partners is an interesting idea, concerns exist about the extent to which the data support this. More detailed points are provided below.

Major points

1. The authors emphasize Nrxn1 and PTPdeltaB- ligand competition as the mechanism through which Neurologin 3 activity is controlled. The data supporting ligand competition are very limited, though, and the in vitro cell-surface binding assay needs to be supported by additional evidence. Can PTPdeltaB- reciprocally displace cell-surface bound Nrxn1? Competitive binding should be demonstrated in the Nlgn3hse and Nlgn3mf mice neurons. Importantly, the authors should demonstrate increased Neurologin 3-Nrxn1 interaction in Nlgn3mf mice in order to conclude that a PTPdeltaB- dependent change in this interaction modulates synapse development.

2. How is excitatory and inhibitory synapse density altered in neurons from the Nlgn3hse and Nlgn3mf mouse lines? This information is required to interpret the data from the bead-based assays of synapse induction. Also, to what extent are excitatory and inhibitory synapse density altered in the Nlgn3hse and Nlgn3mf mouse lines? This should be analyzed in regions relevant for the behavioral tests presented here.

3. Do Neurologin 3 and PTPdelta form a complex in vivo? This should be tested using synaptic fractions. Also, given that Ptprd meB(-) mRNA abundance is less than 5% of total Ptprd mRNA, it is unclear how impactful any competition with Nrxn1 for binding to Neurologin 3 is.

4. The social interaction data in Figure 5e and 5g are not sufficiently compelling to conclude that the *Nlgn3^{hse}* and *Nlgn3^{mf}* mouse lines have different social preferences, or that a possible modest difference is biologically meaningful.

Minor points

5. The authors interpret the data in Fig. 5a-d to suggest that “the canonical and non-canonical pathways seem to counterbalance each other for inhibitory synapse formation”. This seems in conflict with data from Fig. 1 showing that *Nrxn1* induces even more Gephyrin accumulation compared to *PTPdeltaB⁻*.

6. In Fig.6d, only total protein levels for the synaptic markers were analyzed and not their synaptic membrane levels. Such synaptic expression analysis will be necessary to make correlations with the electrophysiological data. Also, data for the synaptic protein levels for *Shank2* and Gephyrin in the *hse* and *mf* mice will need to be shown to interpret the synaptogenic *in vitro* assays.

7. Can the author discuss what partner the more prominent isoform of *PTPdelta* containing splice site B engages to induce synaptic sites? Their data show that it is not Neuroigin 3. Is it *IL1RAPL1/IL-1RAcP*?

8. With respect to the idea that *Nrxn1* and *PTPdeltaB⁻* compete with each other for Neuroigin 3 binding and that this controls social behavior, do the authors have data whether they are they coexpressed in the relevant circuits?

9. Since the R451C mutation reduces Neuroigin 3 expression by ~90% and in turn lowers surface trafficking, and since this mutation likely has dominant negative effects, the analysis of its role in dissecting extracellular synaptic interactions of *Nlgn3* needs justification.

Major points

Responses to the comments from Reviewer #3

1. The authors emphasize *Nrxn1* and *PTPdeltaB⁻* ligand competition as the mechanism through which Neuroigin 3 activity is controlled. The data supporting ligand competition are very limited, though, and the *in vitro* cell-surface binding assay needs to be supported by additional evidence. Can *PTPdeltaB⁻* reciprocally displace cell-surface bound *Nrxn1*? Competitive binding should be demonstrated in the *Nlgn3^{hse}* and *Nlgn3^{mf}* mice neurons. Importantly, the authors should demonstrate increased Neuroigin 3-*Nrxn1* interaction in *Nlgn3^{mf}* mice in order to conclude that a *PTPdeltaB⁻* dependent change in this interaction modulates synapse development.

According to the comment, we carried out following three types of experiments to show PTP δ meB(-)s and NRXNs ligand competition for binding to NLGN3 in vitro and in vivo, and revised the manuscript based on these experimental data.

(1) We performed a complex displacement assay (competitive cell surface binding assay), where PTP δ A3B⁻ displaced cell-surface bound NRXN1 β (Supplementary Fig. 2d; Page 8, Lines 12–14).

(2) In bead-neuron coculture assays, we showed that staining signals for NLGN3 on PTP δ A3B⁻-beads were increased in *Nlgn3^{hse}* neuron cultures compared to in control ones (Fig. 5c; Page 12, Lines 5–6). Conversely, staining signals for NLGN3 on NRXN1-beads were stronger in *Nlgn3^{mf}* neuron cultures than in control ones (Fig. 5d; Page 12, Lines 8–11).

(3) We performed NLGN3-coimmunoprecipitation experiments using synaptosomal fractions of wild-type and *Nlgn3* mutant mice we established, and found that the amount of NLGN3-PTP δ complex was increased in the brain from *Nlgn3^{hse}* mutant mice (Fig. 5e; Page 12, Lines 11–13). Unfortunately, we failed to examine the amount of NLGN3-NRXN complex from the immunoprecipitates because DTSSP used to crosslink protein complexes made NRXNs non-immunoreactive to our NRXN antibodies.

2. How is excitatory and inhibitory synapse density altered in neurons from the Nlgn3hse and Nlgn3mf mouse lines? This information is required to interpret the data from the bead-based assays of synapse induction. Also, to what extent are excitatory and inhibitory synapse density altered in the Nlgn3hse and Nlgn3mf mouse lines? This should be analyzed in regions relevant for the behavioral tests presented here.

We quantified staining signals of both excitatory (Shank2) and inhibitory (Gephyrin) synapse markers in the cultured cerebral cortical neurons used in the beads-neuron co-culture assay and no significant changes were observed in neurons from *Nlgn3^{mf}* and *Nlgn3^{hse}* mice (Figure 1 for the reviewers). We mentioned this fact in the Result section as follows:

Page 12, Lines 2–3, “... These mutations did not seem to change background staining signals for Shank2 and gephyrin in cultured neurons (data not shown).”

We also performed immunohistochemical analyses for VGluT1 and VGAT to estimate excitatory and inhibitory synapse density/balance in our mutant lines (Supplementary Fig. 6h,i; Page 14, Lines 19–21). Although it is difficult to correspond behavioral phenotypes and brain regions, we at least found that relative VGAT staining signals versus VGluT1 signals were stronger in the *Nlgn3^{mf}* line in the medial prefrontal cortex, where some of social phenotypes of *Nlgn3^{R451C}* mutants are shown to be responsible.

Fig.1 for the reviewers: Background Shank2 and gephyrin signals in cultured cortical neurons from *Nlgn3^{hse}* (top) and *Nlgn3^{mf}* mutant mice used in the Fig.5a–d.

3. Do *Neurologin 3* and *PTPdelta* form a complex in vivo? This should be tested using synaptic fractions. Also, given that *Ptprd meB(-)* mRNA abundance is less than 5% of total *Ptprd* mRNA, it is unclear how impactful any competition with *Nrxn1* for binding to *Neurologin 3* is.

As mentioned above in the responses to the major point #1, we performed coimmunoprecipitation analysis with anti-NLGN3 antibody using DTSSP-crosslinked crude synaptosomal (P2') fraction from striatum showing highest *Ptprd A3B-* mRNA level, and demonstrated that NLGN3 and PTP δ actually form a complex in vivo (Fig. 2e and Supplementary Fig. 2c; Page 7, Lines 16–19). However, since we could not examine the amount of NLGN3-NRXN complex in vivo in the *Nlgn3^{mf}* brain as mentioned above, it still remains unclear to what extent the lack of proportionally few PTP δ -meB(-)s affect NLGN3-NRXNs binding and exert phenotypes in vivo. We discussed about possible mechanisms by which lack of proportionally few non-canonical signaling for synaptogenesis exerted robust effects in *Nlgn3^{mf}* mice as follows:

Page 17, Lines 5–8, “The robust increases of inhibitory synaptic proteins in the forebrain of *Nlgn3^{R/C}* or *Nlgn3^{mf}* mutants, despite of proportionally very low contribution of the non-canonical synaptogenic pathway, imply some circuit or system-level mechanisms to compensate for the lack of the non-canonical NLGN3 pathway in these mice.”

4. The social interaction data in Figure 5e and 5g are not sufficiently compelling to conclude that the *Nlgn3^{hse}* and *Nlgn3^{mf}* mouse lines have different social preferences, or that a possible modest difference is biologically meaningful.

We agree with the reviewer that our previous social interaction data are not sufficient enough to show qualitative differences between *Nlgn3^{hse}* and *Nlgn3^{mf}* mice. We thus further analyzed video images during three-chamber sociability test and found that *Nlgn3^{mf}* mice exhibited decreased duration of each sniffing of the stranger mouse, which may support qualitative difference of social behavior between *Nlgn3^{hse}* and *Nlgn3^{mf}* mutant mice (Supplementary Fig. g,h; Page 12, Line 28 – Page 13, Line 3).

Minor points

5. The authors interpret the data in Fig. 5a-d to suggest that “the canonical and non-canonical pathways seem to counterbalance each other for inhibitory synapse formation”. This seems in conflict with data from Fig. 1 showing that *Nrxn1* induces even more *Gephyrin* accumulation compared to *PTPdeltaB-*.

In previous Fig. 5a-d (revised Fig. 5a,b), relative inductions of excitatory and inhibitory synaptogenesis in the *Nlgn3^{hse}* and *Nlgn3^{mf}* neurons compared to in the wild-type neurons are presented (staining signals in the wild-type neurons are set to be 100%) while amounts of synapse induction (intensities of staining signals) are presented in Fig.1b. In fact, absolute staining signal intensity on PTPδA3B-beads and NRXN1β-beads depends on each culture preparation and staining experiment, therefore balancing mechanisms between the canonical and non-canonical NLGN3 pathway should be measured by relative comparisons between normal condition (wild-type neurons) and the condition in which either pathway is disrupted (*mf* or *hse* neurons). Thus, we think it difficult to discuss based on comparison of Fig.1 and Fig. 5.

6. In Fig.6d, only total protein levels for the synaptic markers were analyzed and not their synaptic membrane levels. Such synaptic expression analysis will be necessary to make correlations with the electrophysiological data. Also, data for the synaptic protein levels for *Shank2* and *Gephyrin* in the *hse* and *mf* mice will need to be shown to interpret the synaptogenic *in vitro* assays.

We presented total protein levels in Fig. 6d, intending to compare biochemical phenotypes of our mutants with those of R451C model according to the previous work by Tabuchi et al. (Science 318, 71–76, 2007). We appreciate the reviewer’s suggestion that analysis of protein levels on the synaptic

membrane may support our electrophysiological data and synaptogenic assays in culture system. Due to the limitation of our mutant mice available, we employed immunohistochemical analyses for excitatory and inhibitory synaptic markers, VGluT1 and VGAT, in cortical layer 2/3 and found that relative VGAT/VGluT1 staining signal ratios were increased in the *Nlgn3^{mf}* mutant mice, but not in the *Nlgn3^{hse}* mutants (Supplementary Fig. 6h,i; Page 14, Lines 19–21). We understand the immunostaining signals in the neuropil layer do not necessarily reflect synaptic protein levels, however, these data will support to interpret correlations of our biochemical and electrophysiological data.

7. Can the author discuss what partner the more prominent isoform of PTPdelta containing splice site B engages to induce synaptic sites? Their data show that it is not Neuroligin 3. Is it IL1RAPL1/IL-1RAcP?

Besides IL1RAPL1 and IL-1RAcP, Slitrk1-6 and SALM1-5 have been known as postsynaptic ligands for PTP δ B+ variants. IL1RAPL1/IL-1RAcP–PTP δ interactions are both meA and meB dependent while Slitrks–PTP δ interactions solely depend on meB. SALMs bind to the meB-containing PTP δ variants with higher affinities than to the meB-lacking variants. These complicated combinations and preferences of PTP δ splice inserts as well as differential expressions of PTP δ splice variants and their ligands across brain regions make it difficult to clarify the contribution of each transsynaptic PTP δ complex for synaptic induction. Our results of IL1RAPL1/IL-1RAcP DKO neurons, nonetheless, suggest IL1RAPL1/IL-1RAcP are major ligands for PTP δ A9B+ to induce synaptogenesis at least in our co-culture system of cortical neurons. Therefore, we added following sentence in the Results section.

Page 6, Lines 20–22, “... double-knockout (DKO) mice (Supplementary Fig. 1a,b), suggesting IL1RAPL1 and IL-1RAcP are major postsynaptic ligands for PTP δ A9B+ to induce excitatory synapse formation in the cortical neurons.”

*8. With respect to the idea that *Nrxn1* and *PTPdeltaB-* compete with each other for Neuroligin 3 binding and that this controls social behavior, do the authors have data whether they are they coexpressed in the relevant circuits?*

Our attempt to identify the circuits where PTP δ B–/NRXN competition actually occurs to regulate social behavior has not been going well due to lack of tools to efficiently detect PTP δ B– variants in vivo in the brain, so we now try to obtain PTP δ B– specific antibodies available for immunohistochemical analysis. With respect to the PTP δ B–/NRXN competition for NLGN3 binding, we

found from newly added NLGN3 co-IP experiments that PTP δ -NLGN3 complex is formed more at least in the striatal synaptosomal fraction of the *Nlgn3^{hse}* mice than in that of wild-type littermates (Fig. 5e). This may support the idea that PTP δ B-/NRXN competition/co-expression actually occurs at some synapses within the striatum, where we chose as a target brain region just because of highest *PtprdB*-expression.

9. *Since the R451C mutation reduces Neuroligin 3 expression by ~90% and in turn lowers surface trafficking, and since this mutation likely has dominant negative effects, the analysis of its role in dissecting extracellular synaptic interactions of Nlgn3 needs justification.*

We agree with the comment on the interpretation of the impact of the R451C mutation in the analyses of transsynaptic signaling. Because we could not address experimentally the relations between the trafficking defects/dominant negative effect of the mutant and the changes in the PTP δ and NRXN-mediated synaptogenic pathways, we added discussion about a possible mechanism by which the R451C mutation increases NRXN-mediated synaptogenesis.

Page 16, Lines 24–27, “... The upregulation of NRXN1-mediated inhibitory synaptogenesis in *Nlgn3^{R/C}* neurons, despite of the decreased surface expression and NRXN1 binding affinity of the NLGN3 R451C protein, may suggest some ligand-level compensation mechanism involving other inhibitory postsynaptic NRXN1 ligands such as NLGN2 and Cblns.”

Comments from Reviewer #4:

In this contribution, Yoshida et al. report the identification of protein tyrosine phosphatase δ (PTP δ) as a novel interactor for Neuroligin-3 (NLGN3). They define this interaction as a non-canonical pathway as opposed to the canonical neurexin-neuroligin (NRXN-NLGN) pathway. By introducing structure-guided mutations into NLGN3, they aim to achieve pathway selectivity, suggesting that balanced signalling through both pathways is important for proper development of sociality.

The unbiased discovery of PTP δ using proteomic approaches is convincing and ties in with the previous discovery of new binding partners for NRXN (notably LRRTM and Cbln) and for NLGN (notably MDGA). It is very interesting to see that signalling pathways initiated by Type IIA receptor protein tyrosine phosphatases, an abundant group of presynaptic hub proteins which bind many postsynaptic partners of their own, converge onto the postsynaptic NLGN hub, in this case the autism related NLGN3 specifically.

The work has important implications, is novel and is overall well executed. I support its acceptance if

a number of major and minor issues are first resolved. From my domain of expertise, I will mainly comment on the molecular and structural aspects, as well as the overall biology.

Major remarks;

- 1. The absence of any discussion or experimental inclusion of MDGA as a major regulator of the canonical NRXN-NLGN interaction is remarkable and worrying. Discovered in 2013 and structurally described in 2017 by a trio of papers in Neuron, the MDGA-NLGN interaction has direct implications for this work, as MDGA proteins also compete with NRXN for binding to NLGN, using the same interface now also used by PTP δ . In other words, it is nearly certainly the case that MDGA will also act as a negative regulator for the NLGN3-PTP δ interaction/pathway described in this work. However, this is completely omitted from this work although it would significantly strengthen it further. It should be addressed using competition cell-binding or co-culture assays, by including a structural comparison of binding interfaces, and by including MDGA in the intro/discussion.*
- 2. Spatial and temporal distribution of the NLGN3-PTP δ complex. The cell-based binding assays and bead-cortical neuron co-culture assays convincingly show that PTP δ and NLGN3 have the capacity to bind each other in a splice-dependent manner. However, there is no proof in the form of e.g. a proximity ligation assay or microscopic co-localisation that these two proteins actually meet in vivo in contacting neurons, which should be addressed either in co-cultured neurons or slices. The jump from in vitro cell-based experiments to full-blown in vivo and behavioural experiments is very abrupt, with little consideration for the fundamental synaptic biology of this novel complex.*
- 3. The NLGN3-PTP δ -A3B- complex shows an interesting symmetric W-shaped complex with positions of N- and C-termini compatible with a trans-synaptic binding event. The nominal resolution of the crystallographic data is 4.15Å, which is low. The data have seemingly been processed to a correlation coefficient between random half data sets (CC1/2) cutoff value of ~0.5. I suggest the authors try to squeeze out more data by including weaker reflections beyond this cutoff and by performing paired refinements to see if inclusion of these higher resolution reflections improves density, which can aid interpretation at this resolution.*
- 4. Currently, there is not a single snapshot of the quality of the electron density map in the paper. A 2FoFc density map supplementary figure for the view in Fig. 3a contoured at a meaningful σ value should illustrate the quality of the density and increase confidence in domain placement. Similarly, in Fig. 4 a-d, putative side chain interactions are shown, however it is unlikely that at a resolution of 4.15Å most of them are resolved, let alone that they can be unambiguously placed. Again, corresponding 2FoFc density map supplementary figures must corroborate the accuracy of the model building in these interface regions and the conclusions drawn from it.*
- 5. Related to the above comments, the “Crystallography” subsection within the “Methods” section does not detail how the low-resolution refinement was approached. How was the model restrained/constrained? Has only rigid body refinement been performed, or have side chains been*

positionally refined (which I infer), etc.? A more detailed description of the strategy for the low-resolution data processing, refinement and model building is warranted.

6. ITC analysis between NLGN3 and PTP δ -A3B- interestingly shows an endothermic reaction compensated by large increase of total system entropy. This is in contrast to the NLGN-NRXN interaction which is strongly exothermic by virtue of an interface Ca²⁺ ion. I note that the buffer background signal in almost all ITC experiments is high, often in the order of ~ 0.10 μ cal/sec (for example in Fig. 2g) and often of the same magnitude as the actual protein-protein signal. This may partially obscure the real protein-protein thermodynamic signal. It may be due to their use of the buffer Tris that has a large enthalpy of ionization (HEPES is generally better). Due to the low affinities between NLGN3 and PTP δ and NRXN proteins, the resolution c-value of the ITC thermograms is low, which, combined with the high buffer signal, obscures the inferred differences between WT, MF/IS, and HSE/AAA mutants; I am not convinced by the stated 6-fold reduction in K_d for the MF/IS mutant, based on the

shape of the thermogram. Equilibrium SPR is a better option here for more accurate K_d determination and this complementary approach will strengthen the conclusions, as well as provide binding kinetics. For all ITC measurements, the error on the determination of ΔH , ΔS and K_d should be stated, which is also put out by the Origin analysis software.

7. PTP δ -A3B- MF/IS and MF/AA mutants were designed to specifically block the NLGN-PTP δ -A3B- interaction. The authors go on to use the MF/AA mutant in vivo. However, in Fig. S5 the ITC data for the MF/AA mutant is not shown. I realise the behavior of the MF/IS and MF/AA mutants is similar in the pulldown assay (Fig. 4g), but since the MF/AA mutant is used in vivo the corresponding biophysical data should formally be shown. It is also of note that the MF/IS mutation does actually not fully block binding of PTP δ -A3B- to NLGN3 but indeed still shows an apparent 6-times lower, but measurable K_d of ~ 27 μ M (but see also comment above). It will then be interesting to see whether the MF/AA mutant is the same or blocks the interaction to a larger extent, compatible with the absence of Shank2 and Gephyrin signals (Fig. 5d). In the absence of a full interaction block, it raises questions about any residual signalling activity that may influence the behavioral data.

Minor remarks;

8. The authors should comment on whether the low resolution of the complex dataset may be due to the presence of the PTP δ Fn1 domain, which is very solvent exposed in the structure. Fn1 does not seem to be involved in making any contacts with NLGN3 and the rationale for including it is not clear; the paper does not contain any domain-deletion experiments to first converge on the minimal set of domains necessary for the interaction, which is normal practice. Deletion of Fn1 from the crystallisation construct may lead to a more defined sub-complex and better diffracting crystals.

9. Page 14, Discussion; "In fact, among the PTP δ -meB(-)s, the meA3-containing one showed the highest affinity to NLGN3"; with this cell-surface binding assay in Fig. 2B, true affinity is not really measured. An appropriate biophysical assay such as SPR that directly measures protein-protein

interactions will be more accurate and conclusive, and with good probability show a better correlation with the synaptogenic effects of meA's of varying length.

10. It is well established that NLGN3 forms heterodimers with NLGN1 at excitatory synapses (Poulopoulos et al., 2012). Thus, at these synapses, there is the potential that NLGN1/NLGN3 heterodimers integrate NRX and PTP δ signalling. In that scenario, NRX and PTP δ would act synergistically instead of antagonistically. This hypothesis remains untested and is absent from the author's model, but it should at least be discussed in the light of the experimental findings.

11. The interface analysis focuses on competition of PTP δ with NRXN for binding to NLGN. Of course, there is also another aspect; the potential competition of NLGN with IL1RAPL1/IL1RAC1 for binding to PTP δ ; the latter interaction was structurally described by the authors in 2015. A structural comparison of NLGN vs. IL1RAPL1 for binding to PTP δ should be included with appropriate discussion.

Responses to the comments from Reviewer #4

Major remarks;

1. The absence of any discussion or experimental inclusion of MDGA as a major regulator of the canonical NRXN-NLGN interaction is remarkable and worrying. Discovered in 2013 and structurally described in 2017 by a trio of papers in Neuron, the MDGA-NLGN interaction has direct implications for this work, as MDGA proteins also compete with NRXN for binding to NLGN, using the same interface now also used by PTP δ . In other words, it is nearly certainly the case that MDGA will also act as a negative regulator for the NLGN3-PTP δ interaction/pathway described in this work. However, this is completely omitted from this work although it would significantly strengthen it further. It should be addressed using competition cell-binding or co-culture assays, by including a structural comparison of binding interfaces, and by including MDGA in the intro/discussion.

We examined the effect of MDGA1 on the PTP δ -NLGN3 interaction by competitive cell surface binding assay and found that MDGA1 interfered with PTP δ -NLGN3 interaction. We add a new figures for these results and structural comparison of NLGN1-MDGA1 complex and NLGN3-PTP δ complex (Supplementary Fig. 2f,g; Page 8, Lines 20–22). A possible function of MDGA1 as negative regulator to the NLGN3–PTP δ pathway was discussed in Discussion as follows:

Page 15, Line 19–24, "... Structural comparison between the NLGN3–PTP δ A3B⁻ and NLGN1–MDGA1 complexes shows an overlap of their binding interfaces, and MDGA1 can weakly interfere

with the binding between NLGN3 and PTP δ A3B⁻, as suggested by the competitive cell surface-binding assay mentioned above (Supplementary Fig. 2f). MDGA1 might function as a negative regulator for the PTP δ -meB(-)S-NLGN3 pathway as well as for the NRXNs-NLGN pathway.”

2. Spatial and temporal distribution of the NLGN3-PTP δ complex. The cell-based binding assays and bead-cortical neuron co-culture assays convincingly show that PTP δ and NLGN3 have the capacity to bind each other in a splice-dependent manner. However, there is no proof in the form of e.g. a proximity ligation assay or microscopic co-localisation that these two proteins actually meet in vivo in contacting neurons, which should be addressed either in co-cultured neurons or slices. The jump from in vitro cell-based experiments to full-blown in vivo and behavioural experiments is very abrupt, with little consideration for the fundamental synaptic biology of this novel complex.

We performed following two experiments to detect the NLGN3-PTP δ complex in vivo in the mouse brain and in cultured neurons. We did not adopt slice-based experiments due to lack of PTP δ antibodies available for immunohistochemical studies.

(1) In anti-NLGN3 antibody-mediated co-immunoprecipitation experiments of synaptosomal fraction from striatum, we showed that NLGN3 and PTP δ actually form a complex in vivo (Fig. 2e; Page 7, Lines 16–19).

(2) In bead-neuron coculture assays, we showed that PTP δ A3B⁻-beads induced NLGN3 accumulation on the beads in wild-type neurons but not in the *Nlgn3^{mf}* neurons (Fig. 5d; Page 12, Lines 8–11).

3. The NLGN3-PTP δ -A3B- complex shows an interesting symmetric W-shaped complex with positions of N- and C-termini compatible with a trans-synaptic binding event. The nominal resolution of the crystallographic data is 4.15 Å, which is low. The data have seemingly been processed to a correlation coefficient between random half data sets (CC1/2) cutoff value of ~0.5. I suggest the authors try to squeeze out more data by including weaker reflections beyond this cutoff and by performing paired refinements to see if inclusion of these higher resolution reflections improves density, which can aid interpretation at this resolution.

We processed the diffraction data again to include weaker reflections. The resultant data collection statistics suggested that an efficient resolution could be ranged from 3.7 Å (CC1/2 > ~0.37) to 3.85 Å (I/σI > 1.0). We could refine the atomic model at 3.7 Å, 3.85 Å, and 4.15 Å resolutions with

reasonable R_{free} values, and compared the results by inspection of the $2F_o-F_c$ map. No obvious improvement between 3.7 Å and 3.85 Å could be found, and the atomic model was finally refined at 3.85 Å. The extension of the resolution from 4.15 Å to 3.85 Å seemed to improve several parts of the electron density.

4. Currently, there is not a single snapshot of the quality of the electron density map in the paper. A $2F_oF_c$ density map supplementary figure for the view in Fig. 3a contoured at a meaningful σ value should illustrate the quality of the density and increase confidence in domain placement. Similarly, in Fig. 4 a-d, putative side chain interactions are shown, however it is unlikely that at a resolution of 4.15Å most of them are resolved, let alone that they can be unambiguously placed. Again, corresponding $2F_oF_c$ density map supplementary figures must corroborate the accuracy of the model building in these interface regions and the conclusions drawn from it.

The $2F_o-F_c$ map figures corresponding to the molecular models shown in Fig. 3a and Fig. 4a–4d were shown in Supplementary Fig. 5a and b-e, respectively.

5. Related to the above comments, the “Crystallography” subsection within the “Methods” section does not detail how the low-resolution refinement was approached. How was the model restrained/constrained? Has only rigid body refinement been performed, or have side chains been positionally refined (which I infer), etc.? A more detailed description of the strategy for the low-resolution data processing, refinement and model building is warranted.

Description of the low-resolution refinement in the "**Crystallography**" section was revised as follows:

Page 22, Lines 21–25, "... The final model of the NLGN3 ECD–PTPδ Ig1–Fn1 complex was refined at 3.85 Å to R_{work} and R_{free} of 0.253 and 0.286, respectively. Even in this moderate resolution, positional refinement could be applied without secondary structure restraint or reference structure restraint for the final model. During the initial iterative cycles of model improvement and refinement, reference structure restraint was used to improve the main chain geometry. ..."

6. ITC analysis between NLGN3 and PTPδ-A3B– interestingly shows an endothermic reaction compensated by large increase of total system entropy. This is in contrast to the NLGN-NRXN interaction

which is strongly exothermic by virtue of an interface Ca^{2+} ion. I note that the buffer background signal in almost all ITC experiments is high, often in the order of $\sim 0.10 \mu\text{cal/sec}$ (for example in Fig. 2g) and often of the same magnitude as the actual protein-protein signal. This may partially obscure the real protein-protein thermodynamic signal. It may be due to their use of the buffer Tris that has a large enthalpy of ionization (HEPES is generally better). Due to the low affinities between NLGN3 and PTP δ and NRXN proteins, the resolution c -value of the ITC thermograms is low, which, combined with the high buffer signal, obscures the inferred differences between WT, MF/IS, and HSE/AAA mutants; I am not convinced by the stated 6-fold reduction in K_d for the MF/IS mutant, based on the shape of the thermogram. Equilibrium SPR is a better option here for more accurate K_d determination and this complementary approach will strengthen the conclusions, as well as provide binding kinetics. For all ITC measurements, the error on the determination of ΔH , ΔS and K_d should be stated, which is also put out by the Origin analysis software.

In the all ITC experiments performed in this study, the data from the titration of PTP δ or Nrx1 β to the cell without NLGN3 (*i.e.*, buffer only) were collected as background controls, and subtracted from the data from the titration to the cell with NLGN3 (*i.e.*, buffer containing ligands) to minimize the effect derived from the buffer for the calculation of thermodynamic parameters. All subtracted plots reach a plateau at about 0 kcal mol^{-1} , suggesting that the subtraction was performed in an appropriate manner.

We agree that equilibrium SPR is another option to calculate K_d , but it was not suitable for the analysis of the PTP δ -NLGN3 interaction; our SPR analysis could not detect the meB^- preference of NLGN3, which was clearly shown by other functional data *in vitro* and *in vivo*.

The error on the determination of N , ΔH , and K_d was stated in all ITC data (Fig. 2i, Supplementary Fig. 6a-c) that showed sufficient binding. It seems that the error of ΔS is not put out by the Origin analysis software.

7. PTP δ -A3B- MF/IS and MF/AA mutants were designed to specifically block the NLGN-PTP δ -A3B- interaction. The authors go on to use the MF/AA mutant *in vivo*. However, in Fig. S5 the ITC data for the MF/AA mutant is not shown. I realise the behavior of the MF/IS and MF/AA mutants is similar in the pull-down assay (Fig. 4g), but since the MF/AA mutant is used *in vivo* the corresponding biophysical data should formally be shown. It is also of note that the MF/IS mutation does actually not fully block binding of PTP δ -A3B- to NLGN3 but indeed still shows an apparent 6-times lower, but measurable K_d of $\sim 27 \mu\text{M}$ (but see also comment above). It will then be interesting to see whether the MF/AA mutant is the same or

blocks the interaction to a larger extent, compatible with the absence of Shank2 and Gephyrin signals (Fig. 5d). In the absence of a full interaction block, it raises questions about any residual signalling activity that may influence the behavioral data.

Before we edited the *Nlgn3* gene in mice, we had compared the abilities of MF/IS and MF/AA to block PTP δ interaction in cell surface binding assay and found stronger blockade by MF/AA as shown in the revised Supplementary Fig. 6e. We agree that biophysical methods such as ITC and SPR are useful to calculate Kd values, but cell surface binding assay is sensitive enough to compare binding activities in NLGN3/PTP δ interactions within appropriate range of ligand concentration as indicated by sigmoid correlations between cell surface-bound signals and ligand concentrations eg. in Supplementary Fig. 2d. Actually, as shown in the Supplementary Fig. 6e, interaction-blocking abilities of the NLGN3 mutants correspond well with the Kd values estimated by ITC. So, we decided to generate the MF/AA knock-in mice based on our data of cell surface binding assay.

Regarding the interaction blocking ability of the MF/AA mutation in mouse neurons, we showed in revised Fig. 5b that this mutation completely lacked NLGN3 signals on the PTP δ A3B⁻-beads, suggesting complete interaction block in neuronal systems.

Minor remarks;

8. The authors should comment on whether the low resolution of the complex dataset may be due to the presence of the PTP δ Fn1 domain, which is very solvent exposed in the structure. Fn1 does not seem to be involved in making any contacts with NLGN3 and the rationale for including it is not clear; the paper does not contain any domain-deletion experiments to first converge on the minimal set of domains necessary for the interaction, which is normal practice. Deletion of Fn1 from the crystallisation construct may lead to a more defined sub-complex and better diffracting crystals.

Fn1 is involved in the crystal packing and is indispensable for the crystallization. Moderate resolution (~4 Å or worse) is not so surprising about structures of mammalian receptor-like adhesion complexes, likely due to their structural flexibility.

9. Page 14, Discussion; “In fact, among the PTP δ -meB(-)s, the meA3-containing one showed the highest affinity to NLGN3”; with this cell-surface binding assay in Fig. 2B, true affinity is not really measured. An appropriate biophysical assay such as SPR that directly measures protein-protein interactions will be more accurate and conclusive, and with good probability show a better correlation with the synaptogenic

effects of meA's of varying length.

As pointed out by the reviewer, we did not really measure K_d values for the interactions between NLGN3 and PTPδ splice variants. Therefore, we rephrased the sentence as follows, and added statistical data for the comparison between PTPδA3B⁻ and other variants in the corresponding figure legend (legend to Fig. 2).

Page 15, Lines 9–10, "... In fact, among the PTPδ-meB(-)s, the meA3-containing one showed the strongest NLGN3-binding activity in the CSB assay"

10. It is well established that NLGN3 forms heterodimers with NLGN1 at excitatory synapses (Poulopoulos et al., 2012). Thus, at these synapses, there is the potential that NLGN1/NLGN3 heterodimers integrate NRX and PTPδ signalling. In that scenario, NRX and PTPδ would act synergistically instead of antagonistically. This hypothesis remains untested and is absent from the author's model, but it should at least be discussed in the light of the experimental findings.

We agree the synergistic scenario above may be interesting. It is, however, well-known that ligand (SALM5 and IL1RAPL1)-induced dimerization of PTPδ is prerequisite to elicit intracellular signaling for presynaptic differentiation (Goto-ito et al., 2018, Nat. Commun 9, 269.; Won et al., 2017, Front Mol Neurosci 10,327). Similarly, NRXNs are also known to require NLGN-mediated multimerization for synaptic differentiation (Dean et al., 2003, Nat Neurosci 6,708-16; Lee et al., 2012, J Neurosci 32, 4688-701). Thus, NLGN1/NLGN3 heterodimer may rather interfere with the PTPδ-NLGN3-mediated synaptogenesis. Furthermore, our present results suggest that PTPδ/NRXN signaling competition seems to occur for inhibitory, but not for excitatory, synapse formation (Fig. 5a,b).

11. The interface analysis focuses on competition of PTPδ with NRXN for binding to NLGN. Of course, there is also another aspect; the potential competition of NLGN with IL1RAPL1/IL-1RAcP for binding to PTPδ; the latter interaction was structurally described by the authors in 2015. A structural comparison of NLGN vs. IL1RAPL1 for binding to PTPδ should be included with appropriate discussion.

IL1RAPL1 and IL-1RAcP primarily bind to the B⁺ variants of PTPδ *in vivo*, which cannot bind to NLGN3. IL1RAPL1 and IL-1RAcP is not competitive to NLGN3. To avoid the confusion regarding the preference of the splice variant, we will not compare the complex structures of

IL1RAPL1/IL-1RAcP and NLGN3.

Reviewers' Comments:

Reviewer #2:

Remarks to the Author:

The authors have responded to reviewer criticisms in a very straightforward manner, providing additional data, analysis, and interpretation where necessary.

Reviewer #3:

Remarks to the Author:

The authors have further strengthened this work in the revision and addressed my points. Their new results are convincing and support this study, including the analysis of competition of PTPdeltaB and Nrnx1 for Neuroligin 3 binding, the validation that a PTPdeltaB/ Neuroligin 3 complex exists in the striatum in vivo, and the effects of the Nlgn3 mutations on excitatory and inhibitory synaptic markers in the cortex. The extended behavioral data provide additional insights. Together, this is a very thorough and interesting study of the balance of synapse organizers.
Thomas Biederer

Reviewer #4:

Remarks to the Author:

I have examined the author's point-by-point rebuttal closely. I wish to thank the authors for performing textual and experimental revisions to the manuscript, which improved the work substantially in my opinion. I am satisfied with the response to the points I raised and support the article for publication.

The core finding of this study, the unbiased discovery of PTPdelta(meB-) as novel ligand for NLGN3 is very valuable and will inspire future research avenues into further deciphering the molecular complexity and details of the neurexin-neuroligin signalling pathway.